# Global DNA methylation remodeling during direct reprogramming of fibroblasts to neurons

Chongyuan Luo[1,2†], Qian Yi Lee[3,4†‡], Orly Wapinski[5], Rosa Castanon[1], Joseph R Nery[1], Moritz Mall[3§#], Michael S Kareta[3¶**], Sean M Cullen[6], Margaret A Goodell[6], Howard Y Chang[5,7], Marius Wernig[3*], Joseph R Ecker[1,2*]

[1]Genomic Analysis Laboratory, The Salk Institute for Biological Studies, La Jolla, United States; [2]Howard Hughes Medical Institute, The Salk Institute for Biological Studies, La Jolla, United States; [3]Department of Pathology, Institute for Stem Cell Biology and Regenerative Medicine, Stanford University, Stanford, United States; [4]Department of Bioengineering, Stanford University, Stanford, United States; [5]Center for Personal Dynamic Regulomes, Stanford University, Stanford, United States; [6]Stem Cells and Regenerative Medicine Center, Program in Developmental Biology, Baylor College of Medicine, Houston, United States; [7]Howard Hughes Medical Institute, Stanford University, Stanford, United States

*For correspondence:
wernig@stanford.edu (MW);
ecker@salk.edu (JRE)

†These authors contributed equally to this work

Present address: ‡Laboratory of Metabolic Medicine, Singapore Bioimaging Consortium, Agency for Science, Technology and Research, Singapore, Singapore; §Cell Fate Engineering and Disease Modeling Group, German Cancer Research Center (DKFZ) and DKFZ-ZMBH Alliance, HITBR Hector Institute for Translational Brain Research gGmbH, Heidelberg, Germany; #Central Institute of Mental Health, Medical Faculty Mannheim, Heidelberg University, Mannheim, Germany; ¶Department of Pediatrics, University of South Dakota Sanford School of Medicine, Sioux Falls, United States; **Department of Chemistry and Biochemistry, South Dakota State University, Brookings, United States

Competing interests: The authors declare that no competing interests exist.

**Abstract** Direct reprogramming of fibroblasts to neurons induces widespread cellular and transcriptional reconfiguration. Here, we characterized global epigenomic changes during the direct reprogramming of mouse fibroblasts to neurons using whole-genome base-resolution DNA methylation (mC) sequencing. We found that the pioneer transcription factor Ascl1 alone is sufficient for inducing the uniquely neuronal feature of non-CG methylation (mCH), but co-expression of Brn2 and Mytl1 was required to establish a global mCH pattern reminiscent of mature cortical neurons. Ascl1 alone induced promoter CG methylation (mCG) of fibroblast specific genes, while BAM overexpression additionally targets a competing myogenic program and directs a more faithful conversion to neuronal cells. Ascl1 induces local demethylation at its binding sites. Surprisingly, co-expression with Brn2 and Mytl1 inhibited the ability of Ascl1 to induce demethylation, suggesting a contextual regulation of transcription factor - epigenome interaction. Finally, we found that de novo methylation by DNMT3A is required for efficient neuronal reprogramming.

DOI: https://doi.org/10.7554/eLife.40197.001

## Introduction

Mesoderm originated fibroblast cells can be reprogrammed to ectodermal induced neuronal (iN) cells by the overexpression of proneural transcription factor (TF) Ascl1 (*Chanda et al., 2014*; *Vierbuchen et al., 2010*). Co-expression of Brn2 and Mytl1 with Ascl1 (BAM), enhances reprogramming efficiency by suppressing a competing myogenic program (*Chanda et al., 2014*; *Treutlein et al., 2016*). During reprogramming, Ascl1 acts as a pioneer factor by binding to and opening closed chromatin regions, as well as modulating the binding of Brn2 (*Wapinski et al., 2013*), while Myt1l serves as multi-lineage repressor that maintains neuronal identity (*Mall et al., 2017*). The accumulation of abundant mCH is an epigenomic hallmark unique for mature neurons of mammalian brains, which could not be experimentally induced so far (*Guo et al., 2014*; *Lister et al., 2013*). mCH is recognized by MECP2 and serves critical gene regulatory functions in mature neurons

(*Chen et al., 2015*; *Gabel et al., 2015*; *Guo et al., 2014*; *Stroud et al., 2017*). The reprogramming of fibroblasts to neurons using Ascl1 alone or in combination with Brn2 and Myt1l is a powerful way to interrogate the biochemical function of these TFs and given their prominent role during neural differentiation will also help dissecting their potential regulation of mC remodeling during neuron differentiation. Moreover, the role of DNA methylation in experimental induction of the neuronal lineage by TF reprogramming is not known yet and we currently lack a deep characterization of mC landscape of iN cells. Here we analyzed mC reconfiguration during iN cell reprogramming using whole-genome base-resolution methylome sequencing (MethylC-seq) and functionally perturbed methylation dynamics by genetic inhibition of Dnmt3a.

## Results

### Fully reprogrammed iN cells accumulate abundant non-CG methylation

Base-resolution methylomes were generated for iN cells induced by either Ascl1 alone or all three TFs (BAM) in mouse embryonic fibroblasts (MEFs, *Supplementary file 1*). We analyzed the methylomes of control MEFs, cells at initial (Ascl1 2d) and intermediate stages of reprogramming (Ascl1 5d and BAM 5d) and fully reprogrammed iN cells (Ascl1 22d and BAM 22d). 5d and 22d cells were isolated by fluorescence-activated cell sorting (FACS) using a neuronal TauEGFP reporter. We further generated methylomes for in vitro differentiation of neural progenitor cells (NPC, NPC 7d, NPC 14d and NPC 21d) under comparable culturing conditions. Remarkably, Ascl1 or BAM induced significant global accumulation of the neuronal-enriched non-conventional mCH methylation (*Figure 1A*). mCH level reached an average of 0.28% in Ascl1 22d iN cells and a much greater level (0.57%) in BAM 22d iN cells, which is comparable to the frontal cortex tissue of 2 week old mice (*Lister et al., 2013*). In mature mouse cortical neurons (NeuN+), mCH is strongly enriched in the CAC context and mCAC accounted for approximately 34% of all mCH in mature neurons (*Figure 1—figure supplement 1A*). Similarly, we found neuronal reprogramming from fibroblasts was associated with a progressive establishment of mCAC predominance. In fully reprogrammed Ascl1 or BAM 22d iN cells, mCAC became the most abundant form (26% to 27%) of mCH (*Figure 1—figure supplement 1A*). mCH is inversely correlated with gene expression in mature neurons (*Guo et al., 2014*; *Lister et al., 2013*). mCH was very weakly correlated to gene expression in MEFs or in intermediate reprogramming stages (BAM 5d) but became strongly inversely correlated with gene expression in BAM 22d cells suggesting a similar role of mCH in the brain and iN cells (Spearman r = −0.479, p-value<2.2×10$^{-308}$, *Figure 1—figure supplement 1B*). The accumulation of mCH in iN cells was correlated with a significant elevation of genome-wide mCG level from 72.7% in MEF to 79.5% in Ascl1 22d cells, and 82.5% in BAM 22d cells (*Figure 1A*).

### Reprogramming using BAM factors produces a global non-CG methylation pattern similar to cortical neurons

To evaluate whether iN cells recapitulate the mCH landscape of primary neurons in the brain, we compared iN cells to NPC-derived neurons of various times in culture, mature neurons freshly isolated from the brain, different brain regions as well as non-neural tissues with significant levels of mCH accumulation (global mCH/CH >= 0.1, *Figure 1B*) (*Hon et al., 2013*; *Shen et al., 2012*). Using hierarchical clustering of gene body mCH levels, we found that gene body mCH in mature cortical neurons (Exc, NeuN +and Cortex samples in *Figure 1B*) was more strongly correlated with BAM 22d cells (box 1) than with Ascl1 22d cells (box 2) reflecting the more mature state of BAM iN cells compared to Ascl1-iN cells (*Treutlein et al., 2016*). Accordingly, Ascl1 22d cells correlated more strongly to early reprogramming stages (Ascl1 5d and BAM 5d) and non-neural tissues (box 3,4 in *Figure 1B*). Thus, as observed for gene expression patterns, BAM iN cells displayed a mCH landscape that is more similar to mature cortical neurons (*Treutlein et al., 2016*; *Wapinski et al., 2013*).

We identified genes showing similar or distinct mCH patterns between iN cells and mature neurons using K-means clustering (*Figure 1C and D*; *Figure 1—figure supplement 1C*). We identified a large number (2,098, Cluster 15 and 16) of genes that were strongly depleted of mCH in BAM 22d iN cells but showed medium levels of mCH in immature iN cells and Ascl1 22d iN cells (box in *Figure 1C*). Notably, BAM 22d iN cells and matured cortical neurons showed similar hypo mCH for genes in Cluster 15 and 16. Through correlating with transcriptome analysis of iN cell

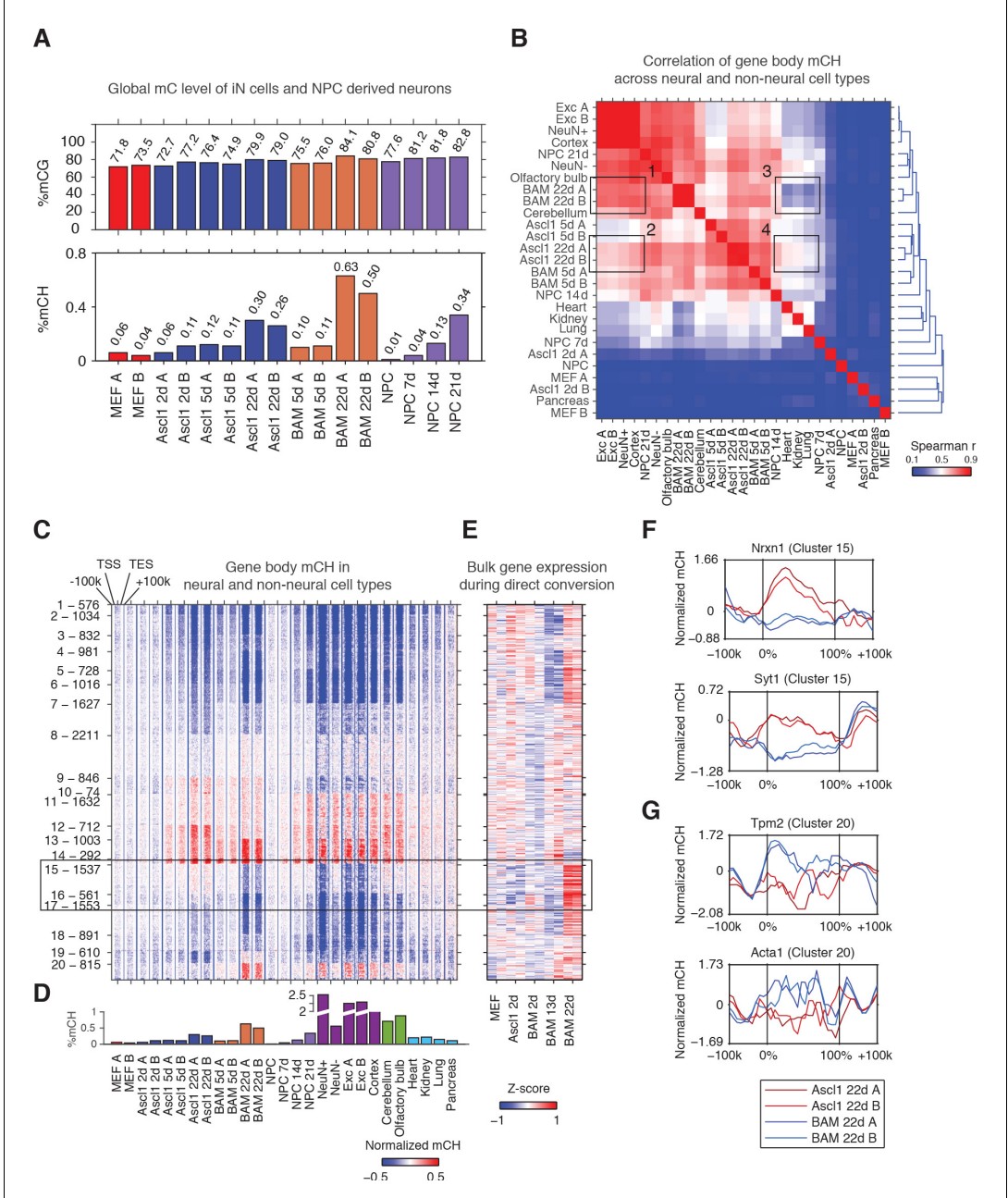

**Figure 1.** The Ascl1, Brn2 and Myt1l reprogramming factors induce an authentic, neuron-specific CH methylation pattern in fibroblasts. (**A**) Global mCG (top panel) and mCH (bottom panel) levels of iN cells and NPC-derived cells. mCG and mCH accumulates in both maturing iN cells and differentiating NPCs. (**B**) Correlation of gene body mCH levels of iN cell reprogramming samples, NPC derived cells, neural and non-neural tissues. Mature cortical neurons clustered more closely to BAM 22d than Ascl1 22d iN cells. NeuN +and NeuN- indicate neuronal and glial nuclei separated using anti-NeuN antibody, respectively (**Lister et al., 2013**). Exc indicates purified nuclei from excitatory neuron expressing Camk2a+ (**Mo et al., 2015**). (**C**) Left panel - K-means clustering of gene body mCH normalized to flanking regions (100 kb surrounding the gene). Reprogramming with BAM produces a global mCH profile more similar to cortical neurons compared to Ascl1 alone. TSS - Transcription Start Site. TES - Transcription End Site. (**D**) Global mCH level of samples compared in (**C**). (**E**) Relative expression (Z-score) of bulk RNA-seq analysis of iN cell reprogramming. Gene expression is inversely correlated to gene body mCH. (**F–G**) mCH pattern at genes with neuronal (**F**) and myogenic functions (**G**) in Ascl1 22d and BAM 22d iN cells. BAM 22d cells show greater depletion of mCH at synaptic genes and accumulation of mCH at myogenic genes, suggesting more efficient activation of neuronal genes and suppression of the alternate myogenic program compare to Ascl1 22d.

DOI: https://doi.org/10.7554/eLife.40197.002

The following figure supplement is available for figure 1:

**Figure supplement 1.** Fully reprogrammed iN cells accumulate CH methylation.

*Figure 1 continued on next page*

*Figure 1 continued*

DOI: https://doi.org/10.7554/eLife.40197.003

reprogramming, we found that genes in Cluster 15 and 16 were actively expressed in successfully reprogrammed iN cells (*Figure 1E* and *Figure 1—figure supplement 1D*) (*Treutlein et al., 2016*). Cluster 15 was enriched in gene functions such as synapse and metal ion binding (*Supplementary file 2*). For example, *Nrxn1* and *Syt1* were depleted of mCH in BAM 22d cells but were enriched of mCH in Ascl1 22d cells (*Figure 1F*). We also found myocyte marker genes *Tpm2* and *Acta1* in Cluster 20, which shows greater level of mCH in BAM 22d iN than Ascl1 22d iN cells (*Figure 1G*). This is consistent with our previous finding that Brn2 and Myt1l can suppress the cryptic myogenic program in iN cell reprogramming induced by Ascl1 (*Treutlein et al., 2016*). In summary, we found direct reprogramming using BAM factors produces a global mCH pattern more similar to cortical neurons, compared to using Ascl1 alone. mCH pattern in BAM iN cells is more permissive for the expression of neuronal and synaptic genes, and more repressive for the expression of the competing myogenic program.

Lastly, we examined the pattern of mCH at long genes in iN cells. It was recently found that long genes are associated with greater levels of mCH in the mouse brain (*Gabel et al., 2015*). Comparing fully programmed iN cells to mouse cortex we found a less pronounced increase in mCH level associated with gene length in iN cells (*Figure 1—figure supplement 1E and F*).

## Non-CG methylation is enriched in dynamically regulated genes during reprogramming and development

To explore the role of mCH in regulating dynamic gene expression during reprogramming, we ranked genes by gene body mCH levels at an early stage of reprogramming (BAM 5d, *Figure 2A–C*). Genes showing early mCH accumulation were strongly enriched in downregulated genes (compared to MEF) in BAM 22d iN cells, and to a less extent enriched in both upregulated and downregulated genes in BAM 13d iN cells (*Figure 2B and C*). Thus early mCH accumulation is correlated with genes showing dynamic expression during reprogramming, and most strikingly with genes repressed in matured iN cells (BAM 22d). We identified up- and down- regulated and static genes during reprogramming by comparing BAM 22d iN cells to MEF, and analyzed mCH accumulation for each gene category across a range of gene expression levels (average expression across reprogramming) (*Figure 2D and E*, *Figure 2—figure supplement 1A and B*). In all expression levels and reprogramming stages examined, downregulated genes accumulated greater levels of mCH than genes with static or increased expression during reprogramming. Surprisingly, we found different patterns depending on the gene expression levels: lowly expressed genes accumulated high levels of mCH regardless of their developmental dynamics (*Figure 2D*; *Figure 2—figure supplement 1A*), whereas for actively expressed genes, gain of mCH is specific to developmentally downregulated genes; the mCH levels of upregulated and static genes were close to the MEF baseline (*Figure 2E* and *Figure 2—figure supplement 1B*). These results suggest a model that mCH is preferentially targeted to two main gene groups - constitutively repressed genes and actively expressed genes showing developmental downregulation.

To test whether mCH is associated with a similar effect on dynamic genes during mouse brain development, we analyzed mCH and gene expression in the cerebellum of mice with conditionally ablated de novo DNA methyltransferase Dnmt3a (Dnmt3a fl/fl;Nes-cre) (*Frank et al., 2015*; *Gabel et al., 2015*; *Hon et al., 2013*). Nes-cre;Dnmt3a fl/fl was shown to erase almost all mCH and significantly reduce mCG level in mouse cerebellum, suggesting DNMT3A is the main DNA methyltransferase responsible for developmental de novo methylation in both CH and CG contexts (*Gabel et al., 2015*). Consistent with our observation in direct reprogramming, mCH level of genes downregulated during cerebellum development (postnatal day 60 vs. day 7) was significantly greater than that of static (Wilcoxon rank sum test, $p=1.2\times10^{-81}$) or upregulated genes ($p=0.0123$, *Figure 2F*). Notably, the removal of mCH and reduction of mCG in the Dnmt3a mutant brain did not affect the expression of developmentally static genes, but resulted in an upregulation of developmentally downregulated genes (*Figure 2G*, $p=1.3\times10^{-12}$), which is consistent with the proposed gene repressive role of mCH. Unexpectedly, Dnmt3a ablation also led to a downregulation of

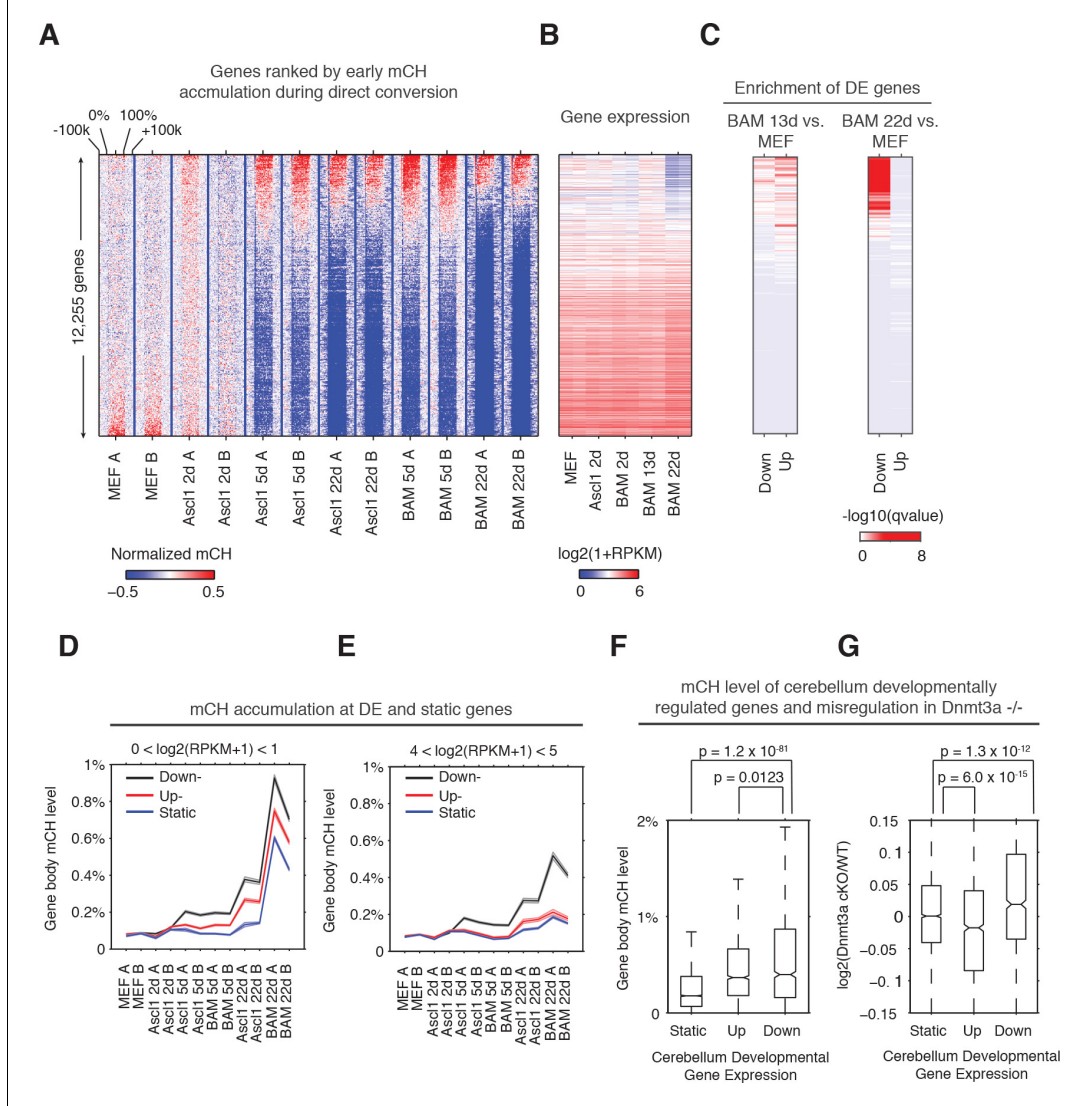

**Figure 2.** Early gene body mCH accumulation predicts later transcriptional downregulation. (**A and B**) Normalized gene body mCH (**A**) and transcript abundance (**B**) for genes ranked by early mCH accumulation at BAM 5d. Early mCH accumulation is strongly correlated to gene repression in BAM 22d iN cells, and both upregulated and downregulated genes in BAM 13d iN cells. (**C**) Significance (hypergeometric test) of the enrichment in down- and up- regulated genes for BAM 13d and BAM 22d iN cells. (**D and E**) Gene body mCH dynamics of static, down- and up- regulated genes with different transcripts abundances - log2(RPKM +1) between 0 and 1 (**D**), between 4 and 5 (**E**) during iN cell reprogramming. (**F**) Gene body mCH level of cerebellum developmentally (P60 vs. P7) regulated genes that were actively expressed (RPKM >3). (**G**) Alteration of cerebellum gene expression by Nes-Cre;Dnmt3a-/- for developmentally regulated genes.

DOI: https://doi.org/10.7554/eLife.40197.004

The following figure supplement is available for figure 2:

**Figure supplement 1.** Dynamically regulated genes are enriched in CH methylation during direct reprogramming and cerebellum development.

DOI: https://doi.org/10.7554/eLife.40197.005

developmentally upregulated genes (p=$6\times10^{-15}$, **Figure 2G**), suggesting that mC support gene expression at these loci. To understand this observation, we first hypothesize that mC antagonizes polycomb repression at developmentally upregulated genes and indirectly promotes gene expression. However, using H3K27me3 peaks (combined from adult cerebellum and P0 hindbrain) as a marker for polycomb regulation, a comparable fraction (17%, p-value=0.87, fisher-exact test) of developmentally up- and down- regulated genes were regulated by polycomb complex. In addition, for either up- or down- regulated genes, genes overlapping with H3K27me3 peak were not affected

differently by Dnmt3a ablation from genes without H3K27me3 (*Figure 2—figure supplement 1C*). Therefore, these results do not support the model that mC indirectly promote gene expression by antagonizing polycomb repression.

We considered a second hypothesis that Dnmt3a indirectly supports developmental gene activation by providing the substrate for hydroxymethylcytosine in a CG dinucleotide context (hmCG). hmCG is associated with actively expressed genes in mouse brain cortex or cerebellum (*Lister et al., 2013*; *Mellén et al., 2017*). In support of this speculation, in adult mouse cerebellum, developmentally upregulated genes showed significantly higher level of gene body hmCG than developmentally downregulated genes (p=$1.3 \times 10^{-6}$, Wilcoxon rank sum test) and static genes (p=$2.4 \times 10^{-55}$, Wilcoxon rank sum test) (*Figure 2—figure supplement 1D*). Developmentally upregulated and downregulated genes show comparable levels of other cytosine modifications (mCG, mCH or hmCH). The result suggests that the activation of developmentally upregulated genes is associated with hmCG accumulation, which may indirectly require DNMT3A for contributing the substrate (mCG) for TET methylcytosine dioxygenases to produce hmCG.

## Promoter CG methylation targets non-neural transcriptional programs during reprogramming

Next, we wanted to determine the role of mCG dynamics in reprogramming. To examine the interaction between promoter (±500 bp from annotated transcription start sites (TSS)) mCG dynamics and gene expression, we ranked 7427 differentially expressed (FDR < 0.05, Fold change >= 2) genes by fold change between MEF and BAM 22d iN cells. We found a pronounced hyper CG methylation signature at the promoter of genes that become repressed during reprogramming (*Figure 3B and C*), such as the *Col1a2* locus (*Figure 3A*). Moreover, *Col1a2* TSS is heavily methylated in NPC and NPC derived neurons, but not in in vivo grown cells such as fetal cortex or mature cortical neurons (NeuN+) or glia (NeuN-), suggesting direct reprogramming remodeled the methylation state of *Col1a2* promoter to that reminiscent in vitro grown neurons (*Figure 3A*). To compare promoter mCG between direct reprogrammed iN cells, in vitro and in vivo grown cells, we identified 4496 promoters that became significantly hyper CG methylated in BAM 22d iN cells compared to MEFs (*Figure 3—figure supplement 1A*). Promoters that became hyper CG methylated during direct reprogramming showed significantly greater mCG in BAM 22d iN cells than in in vivo grown cells (p<$2.9 \times 10^{-7}$, Wilcoxon rank sum test, *Figure 3—figure supplement 1B*), suggesting promoter hyper CG methylation is an epigenomic signature of direct reprogramming. During reprogramming, mCG at promoters showed a strong inverse correlation with gene expression (*Figure 3D*). This negative correlation between promoter mCG and gene expression in iN cells attenuates rapidly within 2 kb of the TSS, which is strikingly different from primary human or mouse tissues (*Figure 3E* and *Figure 3—figure supplement 1C*). Pairwise comparisons of brain cortex to kidney, cerebellum, heart and lung supported the previous analysis of human tissues that the peak negative correlation between mCG and gene expression was found at 2–3 kb downstream of TSS (*Figure 3E* and *Figure 3—figure supplement 1C*) (*Schultz et al., 2015*). Across a broad range gene expression levels, we could identify a subset of genes clearly showing downregulation and promoter hyper CG methylation in BAM 22d iN cells compared to MEF (*Figure 3—figure supplement 1D*). This subset of genes were also more actively expressed in MEF than in adult mouse cortex (*Figure 3F and G*). Therefore the downregulation of genes with hyper CG methylated promoter during direct programming is consistent with their repressed expression in in vivo grown neurons. Consistent with their active expressions in MEF, genes with promoter hyper CG methylation induced during reprogramming were strongly enriched in fibroblast specific gene ontology terms such as extracellular matrix or collagen (*Figure 3H*).

We further asked whether the suppression of myogenic program by BAM factors involves the repression of key gene expression by DNA methylation. Indeed, promoters of a significant fraction (8/15) of analyzed myocytic marker genes (*Cryab*, *Hspb8*, *Mybpc1*, *Tnni1*, *Pgam2*, *Tpm1*, *Tpm2* and *Tnnt2*) were hyper-methylated in BAM 22d and NPC derived neurons but not in Ascl1 22d iN (*Figure 3I and J*) (*Treutlein et al., 2016*). This result correlates with previous data showing repressed myocyte-specific genes in BAM iN cells (*Treutlein et al., 2016*). Only a subset of myocytic marker genes (*Myl1*, *Tnni1* and *Acta1*) were also hyper CG methylated in cortical neurons, suggesting the promoter hyper-methylation at myocytic genes is in part an epigenetic program specific for in vitro grown neurons. We further grouped promoters into three types based on their methylation states in

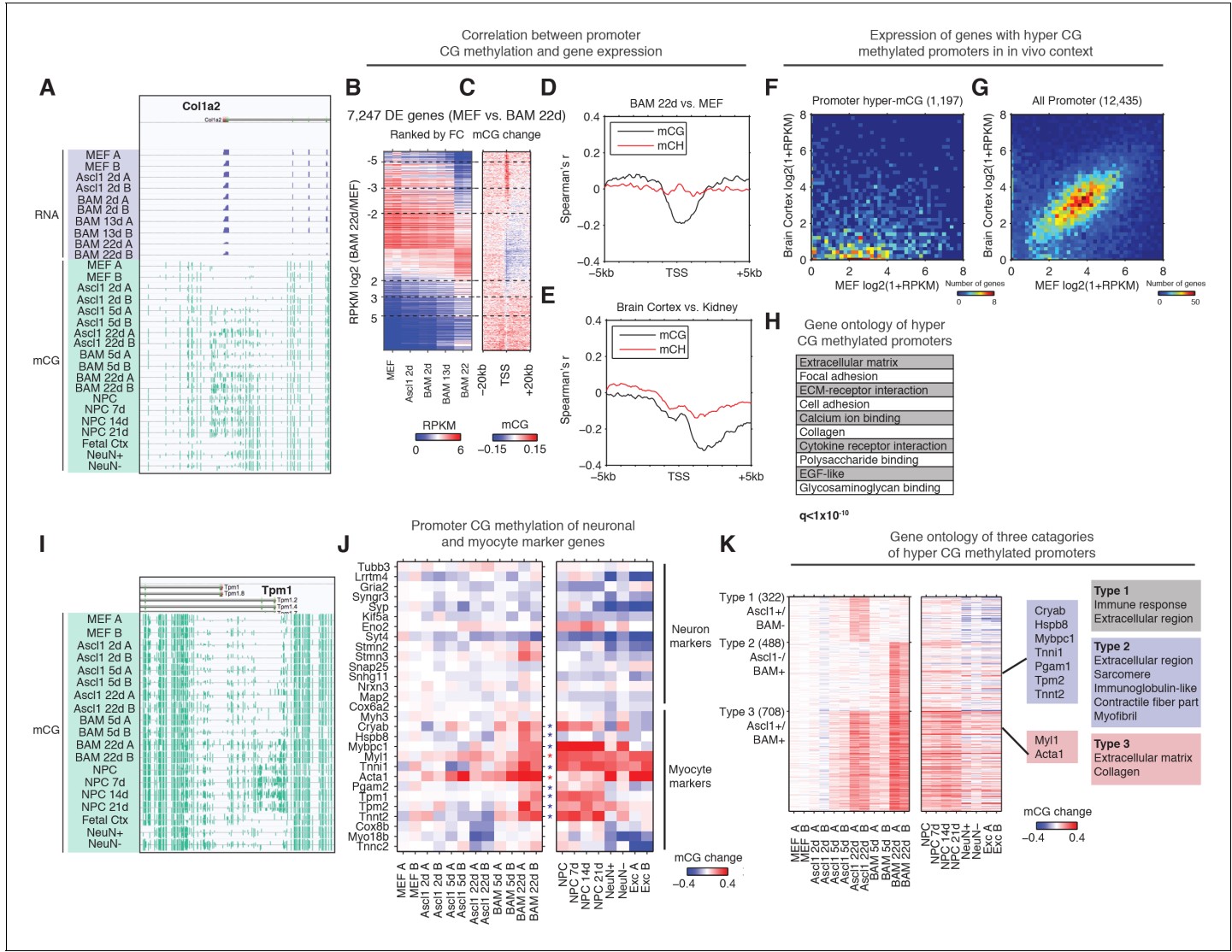

**Figure 3.** Silencing of key non-neuronal programs is associated with promoter hyper CG methylation. (**A**) Browser view of fibroblast marker gene *Col1a2* promoter methylation dynamics and corresponding gene expression during reprogramming. The promoter is strongly methylated throughout NPC differentiation, but is unmethylated in fetal cortex, cortical neurons and glia. However, it gradually accumulates mCG through the course of reprogramming using either Ascl1 alone or BAM factors. (**B**) Differentially expressed genes between BAM 22d iN and MEF were ranked by their relative fold-changes. (**C**) mCG change between MEF and BAM 22d in ±20 kb regions surrounding TSS of differentially expressed genes. Hyper CG methylation was found at TSS for downregulated genes. (**D**) Correlation between mCG change and differential gene expression between BAM 22d iN and MEF in ±5 kb region surrounding TSS. (**E**) Correlation between mCG change and differential gene expression between brain cortex and kidney in ±5 kb regions surrounding TSS. (**F–G**) Scatter plots comparing the expression of genes with hyper CG methylation promoter in BAM 22d iN cells (**F**) or all genes (**G**) between MEF and brain cortex. Hyper CG methylation genes in BAM 22d cells are strongly repressed compared to all genes in the in vivo context. (**H**) Gene ontology term enrichment of genes with hyper CG methylation promoters in BAM 22d iN cells. (**I**) Browser view of promoter mCG dynamics of myocyte marker gene *Tpm1*. The promoter is methylated through NPC differentiation, but is unmethylated in fetal cortex and cortical neurons and glia. However, it remains unmethylated when reprogramming with Ascl1 alone, and only accumulates mCG in the presence of Brn2 and Myt1l in BAM. (**J**) Promoter mCG dynamics of neuron and myocyte marker genes during reprogramming. Blue asterisks indicate promoters hyper-methylated in BAM 22d iN, but not in Ascl1 22d iN. Red asterisks indicate promoters hyper-methylated in both BAM 22d iN and Ascl1 22d iN. A majority of myocyte genes are only hyper methylated by BAM and not Ascl1 alone. (**K**) Left: Heatmaps showing promoter mCG during neuronal reprogramming, NPC differentiation and in primary neurons. Promoters were categorized into three groups showing hyper-mCG (mCG increase >0.1) in Ascl1 22d only, BAM 22d only or both type of iN cells. Right: Gene ontology enrichment of promoters for the three groups. Promoters of fibroblast genes were hyper CG methylated in both Ascl1 alone and BAM iN cells, but only BAM 22d cells exhibit strong hyper CG methylation at myogenic genes.

DOI: https://doi.org/10.7554/eLife.40197.006

The following figure supplement is available for figure 3:

*Figure 3 continued on next page*

*Figure 3 continued*

**Figure supplement 1.** Direct reprogramming induces distinct promoter CG methylation accumulation.

DOI: https://doi.org/10.7554/eLife.40197.007

Ascl1 22d or BAM 22d iN cells (*Figure 3K*). Type two promoters that were specifically hyper CG methylated in BAM 22d cells but not Ascl1 22d, and were enriched in skeletal muscle related function such as sarcomere, contractile fiber part and myofibril (*Figure 3K*). Remarkably, promoters showing hyper CG methylation during reprogramming were marked with elevated H3K27me3 levels in cortical neurons (p<1.2 $\times$ 10$^{-50}$, rank sum test, *Figure 3—figure supplement 1E*), suggesting these loci are repressed with polycomb repressive complexes (PRC) in in vivo grown neurons. Promoter hyper CG methylation of fibroblast and myocyte specific genes may reflect an alternative repressive mechanism (DNA methylation vs. PRC) being utilized during direct reprogramming or cell culture. Taken together, the suppression of fibroblast and myogenic programs during reprogramming was associated with promoter hyper CG methylation, which indicates the potential role of DNA methylation in mediating the suppression of historical and competing programs during reprogramming.

## Reprogramming using Ascl1-alone or BAM factors induce distinct local DNA methylation remodeling

Discrete regions with low levels of DNA methylation have previously been found to indicate TF binding (*Mo et al., 2015*; *Schultz et al., 2015*; *Stadler et al., 2011*). To explore the interaction between CG methylation and TF binding, in particular the binding of ASCL1, BRN2 and MYT1L, we identified 10,075 and 15,093 differentially methylated sites (DMSs) showing reduced mCG during Ascl1-only and BAM induced reprogramming, respectively (*Figure 4A and C*; *Supplementary file 3*). 96% of Ascl1-only DMSs and 95.7% of BAM DMSs were located more than 2.5 kb away from the closest TSS. Thus the vastly majority of DMSs are likely to be associated with distal gene regulatory activities. DMSs were grouped based upon their demethylation kinetics (*Figure 4A and C*). To understand the relationship between DMSs and the binding of TFs that drive the reprogramming of fibroblasts to neurons, we overlapped DMSs and ChIP-seq peaks of ASCL1, BRN2 and MYTL1 requiring a minimum of one base overlap. We compared DMSs with ChIP-seq peaks of ASCL1 following 2 days of overexpression of Ascl1 alone (Ascl1 MEF peaks) or with Brn2 and Myt1l (Ascl1 BAM peaks) in MEF (*Figure 4B and D*) (*Wapinski et al., 2013*). Similarly, DMSs were compared to BRN2 peaks following 2 days of overexpression of Brn2 alone (Brn2 MEF peaks), or with Ascl1 and Myt1l (Brn2 BAM peaks) in MEF (*Figure 4B and D*) (*Wapinski et al., 2013*). The reduction of local mCG during the reprogramming of fibroblasts to iN cells is likely mediated by direct demethylation instead of passive dilution, since we have previously shown that cells exit cell cycle within 2 days of Ascl1 induction based on significant reduction in Ki67 positive cells (*Davila et al., 2013*) and homogenous downregulation of cell cycle genes from single cell RNA-seq data (*Treutlein et al., 2016*). Among the sites that become demethylated in Ascl1-only reprogramming we found a striking co-enrichment of ASCL1 binding, in particular at early demethylating sites (*Figure 4B*; *Figure 4—figure supplement 1C*), suggesting that ASCL1 can recruit a DNA demethylation machinery to its target sites. This fits well with Ascl1's known function as a strong transcriptional activator (*Castro et al., 2011*; *Raposo et al., 2015*; *Wapinski et al., 2013*). Surprisingly, however, only 0.66% (99/15,093) of BAM DMSs were overlapped with ASCL1 peaks (*Figure 4D*; *Figure 4—figure supplement 1C*). DMSs demethylated in early stage of BAM induced reprogramming (BAM DMS group 3) are enriched in BRN2 binding sites in MEF (*Figure 4D*). BAM DMSs demethylated in fully reprogrammed BAM 22d iN cells showed a moderate overlap with BRN2 binding sites in neural stem cells (NSC, *Figure 4D*). BRN2 binding is known to be dependent on existing open chromatin and pioneer factors such as ASCL1 (*Wapinski et al., 2013*). This result suggests BRN2 is able to at least partially access its physiological binding sites in fully reprogrammed BAM 22d iN cells. Although Myt1l serves as multi-lineage repressor during reprogramming, we found only 0.3% (38/15,093) BAM DMSs overlapped with MYT1L binding sites (*Figure 4—figure supplement 1A–C*). Therefore MYT1L binding does not majorly contribute to mCG remodeling during the direct reprogramming.

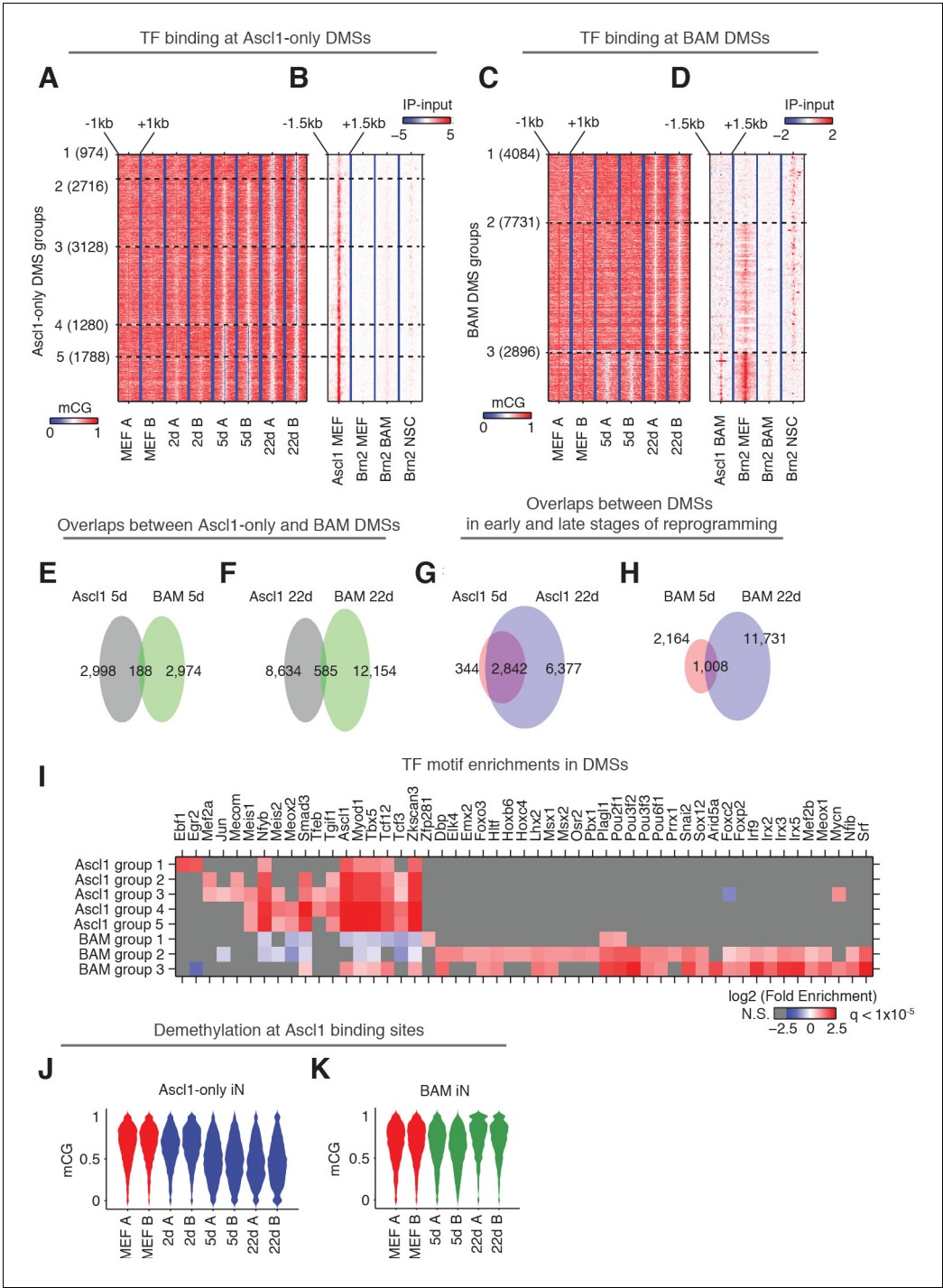

**Figure 4.** Ascl1 and BAM factors induce distinct CG methylation reconfigurations. (**A and C**), DMSs identified and combined from all pairwise comparisons during reprogramming driven by Ascl1 alone (**A**) or BAM (**C**) were ordered by the kinetics of mCG remodeling (see Materials and methods). The plots show clusters containing more than 100 DMSs, which are statistically more robust. (**B and D**) Signal intensity of Ascl1 and Brn2 ChIP-seq were plotted for 3 kb regions surrounding the DMSs. (**E–H**) The overlaps of DMSs identified at different stages of reprogramming driven by Ascl1 or BAM expression. (**I**) TF binding motif enrichment in DMS groups shown in (**A**) and (**C**). Insignificant enrichments (q > $1\times10^{-5}$) were shown as gray. (**J**) The distribution of mCG level at Ascl1 peaks during Ascl1-only induced cells during reprogramming. (**K**) The distribution of mCG level at Ascl1 peaks in BAM induced cells during reprogramming.

DOI: https://doi.org/10.7554/eLife.40197.008

*Figure 4 continued on next page*

*Figure 4 continued*

The following figure supplement is available for figure 4:

**Figure supplement 1.** Differentially methylated sites (DMSa) induced by direct reprogramming overlap with ASCL1 and BRN2, but not MYT1L binding sites.

DOI: https://doi.org/10.7554/eLife.40197.009

Consistent with the drastically different representations of ASCL1 binding sites in Ascl1-only DMSs and BAM DMSs, less than 10% DMSs were shared between Ascl1-only and BAM induced reprogramming (*Figure 4E and F*). In contrast, most DMSs found in Ascl1 5d (89.2%) were also found in Ascl1 22d (*Figure 4G*), suggesting the little overlap between DMSs demethylated during Ascl1-only and BAM induced reprogramming reflects true biological differences. To further illuminate the nature of the demethylated regions we performed motif enrichment analysis for the surrounding 500 bp (±250 bp) regions of each group of DMSs shown in *Figure 4A and C*. As expected, the Ascl1 binding motif was strongly enriched in Ascl1-only DMSs (*Figure 4I*). The DMSs found in the two types of iN cell reprogramming were enriched in completely different TF binding motifs (*Figure 4I*). Accordingly, the Ascl1 motif was only partly enriched in BAM DMSs demethylated in early reprogramming (BAM DMS group 3), but was depleted in BAM DMSs emerged in later stages of reprogramming (BAM DMS group 1). As expected, the Brn2 (Pou3f2) motif along with several other Pou-Homeodomain motifs were enriched in BAM DMSs but not in any Ascl1-only DMSs. Moreover, we found enrichment of additional motifs of TFs many of which with prominent neuronal function such as Lhx2, Emx2, Nfib and Mef2 in BAM DMSs. These findings indicate that the dynamic DMSs during iN cell reprogramming are enriched in DNA regulatory elements and their proper activation involves DNA demethylation as part of the chromatin remodeling of a fibroblast to a neuronal configuration.

Given the striking enrichment of ASCL1 binding at Ascl1-only DMSs, we next analyzed the mCG dynamics of just the Ascl1 binding sites during reprogramming. Although only a moderate 28.4% (935/3296) of Ascl1 binding sites (not already showing lowly methylation in MEF) were overlapped with Ascl1-only DMSs, our quantitative analysis found that Ascl1 expression in fibroblasts induced a substantial reduction of DNA methylation at the vast majority of Ascl1 binding sites in early reprogramming Ascl1 5d with persistence of reduced mCG levels to 22 days (*Figure 4J*). During BAM-induced reprogramming, the ASCL1-bound sites behaved completely differently. Early in the reprogramming process Ascl1 did induce demethylation at many of its target sites, but to a much more moderate degree (BAM 5d, *Figure 4K*). Surprisingly, these changes were only transient, as the demethylation of Ascl1 binding sites was reversed in the mature stage of BAM iN reprogramming (BAM 22d). The results suggest that Brn2 and Myt1l functionally interfere with Ascl1-mediated local demethylation. Since ASCL1 chromatin binding is unaffected by other transcription factors (*Wapinski et al., 2013*), this observation indicates that addition of Brn2 and Myt1l modifies ASCL1's ability to induce DNA demethylation at its chromatin binding sites and that the two reprogramming strategies are associated with drastically different DNA methylation remodeling (*Figure 4J and K*). Such modulation of ASCL1 by BRN2 is supported by the co-binding of the two TFs at about about a quarter of ASCL1 sites in MEF (*Wapinski et al., 2013*).

## Dnmt3a is functionally required for efficient neuronal reprogramming

Next, we assessed the expression of known DNA methylation regulators throughout the entire reprogramming process. We found a strong upregulation of the de novo DNA methyltransferase Dnmt3a, the TET methylcytosine dioxygenase 3 (Tet3) involved in DNA demethylation and the mCH reader Mecp2 expressions during reprogramming (*Figure 5A*) (*Chen et al., 2015*; *Gabel et al., 2015*; *Guo et al., 2014*). The expression of Uhrf1 that encodes a SRA domain protein required for DNMT1-mediated mCG was drastically reduced in BAM 13d and 22d (*Figure 5A*) (*Bostick et al., 2007*), suggesting that the maintenance DNA methylation activity became repressed in post-mitotic iN cells. The other DNA methylation regulators were not dynamically regulating during the reprogramming process. Of the regulated genes, we found that ASCL1 directly binds to Dnmt3a promoter. ASCL1 occupies a binding site 1,957 bp downstream of Dnmt3a transcription start site (TSS) in MEF and initial reprogrammed cells (*Figure 5B*). However, Ascl1 over-expression is insufficient to activate

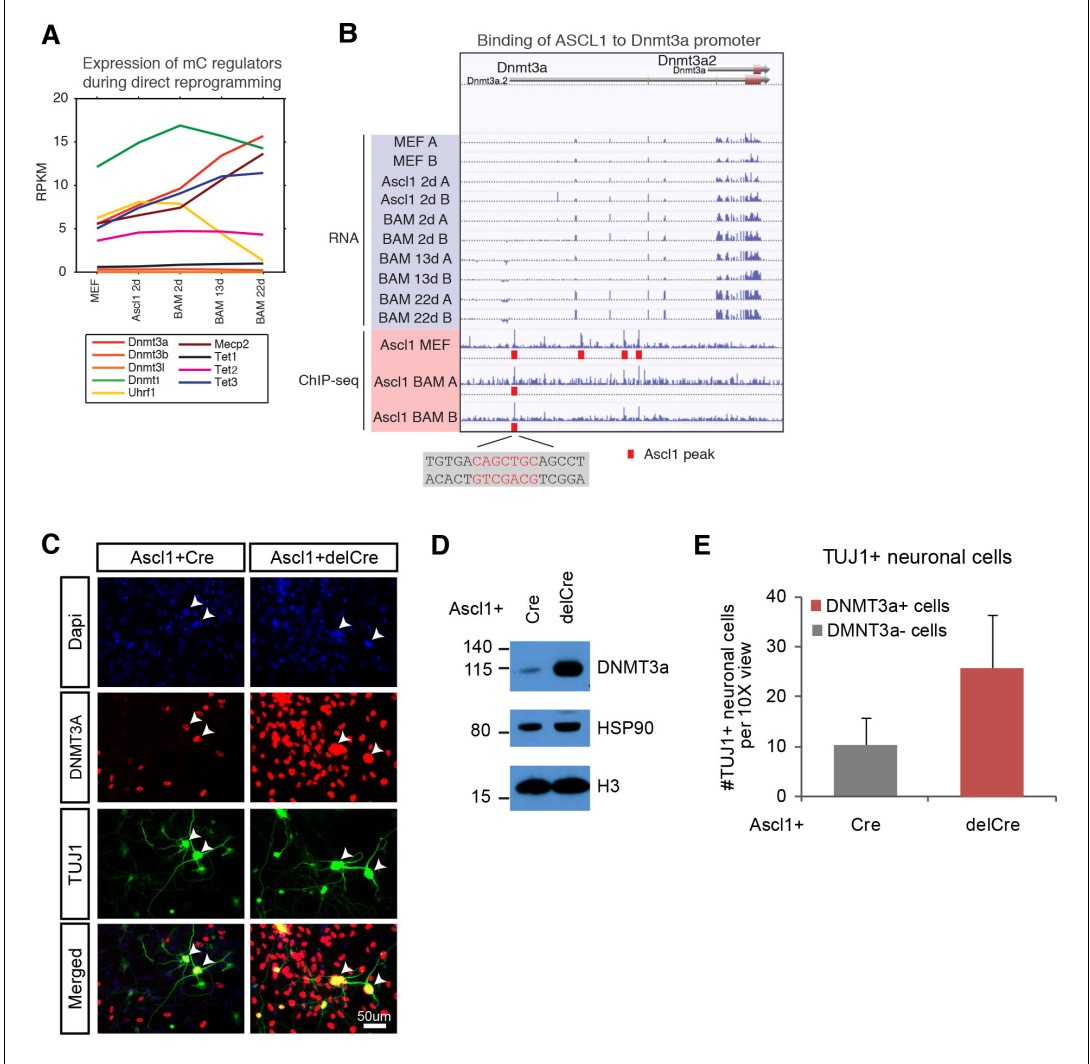

**Figure 5.** de novo DNA methylation is required for efficient reprogramming from fibroblasts to neurons. (**A**) Expression of DNA methylation regulators and readers during neuronal reprogramming. (**B**) Ascl1 ChIP-seq peak is located proximal to Dnmt3a promoter. (**C**) Immunostaining of Dnmt3a/3b[fl/fl] cells 14d post-Ascl1 induction with dox. There appears to be a similar reprogramming efficiency when comparing the CreGfp (left) and delCreGfp (right) conditions. However, most TUJ1 +iN cells in the CreGfp condition were still expressing low levels of DNMT3A (white arrows) despite co-infection with CreGfp. (**D**) Western blot showing efficient knock-out of DNMT3A in Ascl1 +Cre expressing cells compared to Ascl1 +delCre control 13 days post induction of Ascl1. (**E**) Average counts of TUJ1 +iN cells per 10X field of view 14d post-Ascl1 induction that are co-infected with 1) CreGfp and Dnmt3a- (gray) or 2) delCreGfp and Dnmt3a+ (red) (error bars are stderr, n = 3).

DOI: https://doi.org/10.7554/eLife.40197.010

The following figure supplement is available for figure 5:

**Figure supplement 1.** Deletion of Dnmt3a/3b in fibroblasts leads to reduced reprogramming efficiency.

DOI: https://doi.org/10.7554/eLife.40197.011

Dnmt3a expression since Dnmt3a was not significantly upregulated 2 days after Ascl1 induction. It remains unclear whether the binding of ASCL1 to the Dnmt3a promoter functionally contributes to Dnmt3a activation during reprogramming.

To functionally query the role of de novo DNA methylation during reprogramming, we then acutely ablated the two main de novo DNA methylases by co-infecting fibroblasts isolated from Dnmt3a/3b[fl/fl] mice with lentivirus expressing intact Cre recombinase (Cre) or a non-functional truncated form of Cre recombinase (delCre). To initiate reprogramming we co-infected the cells with a doxycycline (dox)-inducible TetO-Ascl1 lentivirus. Dox was added 2–4 days post-infection to allow for degradation of Dnmt3a mRNA and protein before inducing Ascl1 expression for direct

conversion of MEFs to neurons. Dnmt3b is not expressed through the course of reprogramming (*Figure 5A*). Cells were fixed and immunostained with the neuronal marker TUJ1 and DNMT3A (*Figure 5C*; *Figure 5—figure supplement 1A*) 14 days after Ascl1 induction. Despite efficient Dnmt3a depletion in the total fibroblast population (*Figure 5D*; *Figure 5—figure supplement 1C*) we observed a number of Dnmt3a-positive iN cells in the Cre-infected group, presumably due to incomplete degradation of Dnmt3a or escape from Cre recombination (*Figure 5C*; *Figure 5—figure supplement 1D–F*). We therefore considered only the Dnmt3a-negative iN cells in the Cre-treated group. As shown in *Figure 5F*, the reprogramming efficiency of Dnmt3a-negative cells (gray) was reduced compared to control-infected, Dnmt3-positive cells (red), indicating that de novo DNA methylation plays a role in efficient iN cell generation from fibroblasts.

## Discussion

Our study found that fully reprogrammed iN cells accumulate abundant mCH, which is an epigenomic signature of post-mitotic neurons in the mammalian brain (*Guo et al., 2014*; *Lister et al., 2013*). To our knowledge, high level (e.g. mCH/CH > 0.1%) has not been previously reported in any in vitro neuronal differentiation or direct reprogramming models. Robust mCH accumulation suggests that iN cells recapitulates a major epigenomic signature that starts to emerge during the second postnatal week of mouse brain development and supports the notion that fully reprogrammed iN cells resemble differentiated neurons (*Lister et al., 2013*; *Vierbuchen et al., 2010*). The observation that BAM induced iN cells further acquired a global mCH landscape reminiscent to mature neurons is consistent with previous findings that BRN2 and MYTL1 can enhance neuronal maturation during reprogramming (*Chanda et al., 2014*). The reprogramming of fibroblasts to neurons and induction of mCH by overexpressing merely 1 or 3 TFs provides a tractable system to study the dynamic regulation of genome-wide DNA methylation. For example, mCH accumulation during reprogramming is associated with the upregulation of Dnmt3a and may be regulated by the direct binding of ASCL1 to Dnmt3a promoter. In addition, inducible expression of Ascl1 allowed us to conclude that ASCL1 induces local demethylation at most of its binding sites. mCH is inversely correlated with gene expression in the brain and has been shown to mediate gene repression through MECP2 (*Chen et al., 2015*; *Gabel et al., 2015*; *Lister et al., 2013*; *Stroud et al., 2017*). Although the timing of mCH accumulation parallels neuron maturation both in vivo during in vitro reprogramming, little is known whether mCH plays any role in regulating brain development. Our integrative analysis of transcriptomic and DNA methylome datasets for neuronal reprogramming and published work on cerebellar development shows that mCH is enriched in genes that are downregulated during development (*Frank et al., 2015*), suggesting that mCH plays a role in gene repression during brain development. Reanalysis of the Dnmt3a conditional knock-out transcriptome data supported our model - developmental downregulated genes become activated in the absence of mCH (*Gabel et al., 2015*). In both mouse and human, mCH accumulation start around birth and plateaus at early adolescence. Our results collectively suggest that mCH facilitates neuronal maturation during development through gene repression.

Direct conversion from fibroblasts to neurons is associated with distinct promoter hypermethylation, which is unique to iN cells and NPC differentiated cells and is not observed during in vivo neuron differentiation. We thus found iN cells exhibit both consistent (e.g. mCH) and different (e.g. promoter methylation) epigenomic features compared to mature cortical neurons. Our study suggests that de novo DNA methylation contributes to cellular reprogramming by suppressing the donor fibroblast and the competing myogenic programs. The promoters targeted by hyper methylation is an independent set from ones bound by multi-lineage suppressor Mylt1l. In summary, our genomic analysis suggests both types of DNA methylation - mCH and mCG serve repressive functions during cellular reprogramming and primarily target gene bodies and promoters, respectively. This hypothesis is supported by our functional experiment showing that fibroblast cells with ablated Dnmt3a locus have reduced reprogramming potential, which supports hypothesis that de novo DNA methylation is necessary for efficient reprogramming.

# Materials and methods

## Key resources table

| Reagent type (species) or resource | Designation | Source or reference | Identifiers | Additional information |
|---|---|---|---|---|
| Antibody | Mouse monoclonal anti-TUBB3 (Tuj1) | Covance | MMS-435P; RRID:AB_2313773 | (1:1000) IF |
| Antibody | Rabbit polyclonal anti-DNMT3A (H-295) | Santa Cruz Biotech | sc-20703; RRID:AB_2093990 | (1:1000) WB, (1:200) IF |
| Antibody | Chicken polyclonal anti-GFP | Abcam | ab13970; RRID:AB_300798 | (1:1000) IF |
| Antibody | rabbit polyclonal anti-HSP90 | Cell signaling | 4874; RRID:AB_2121214 | (1:1000) WB |
| Antibody | rabbit polyclonal anti-H3 | Cell signaling | 9712 | (1:10000) WB |
| Other | Dapi | ThermoFisher Scientific | D1306 | 1 ug/ml |
| Genetic reagent (*M. musculus*, female) | C57BL/6J | The Jackson Laboratory | 000664; RRID:IMSR_JAX:000664 | |
| Genetic reagent (*M. musculus*, male) | Mapt-tm1(EGFP)Klt | The Jackson Laboratory | 004779 | Homozygous TauEGFP |
| Strain, strain background (M. musculus) | Dnmt3a fl/fl | PMID: 15215868, 25130491,15757890 | | |
| Strain, strain background (M. musculus) | Dnmt3b fl/fl | PMID: 15215868, 25130491,15757890 | | |
| Commercial assay or kit | DNeasy Blood and Tissue Kit | Qiagen | 69506 | |
| Commercial assay or kit | MethylCode Bisulfite Conversion Kit | Invitrogen | MECOV50 | |
| Software, algorithm | Methylpy | PMID: 26030523 | | |
| Software, algorithm | Bowtie2 | PMID: 22388286 | RRID:SCR_016368 | |
| Software, algorithm | STAR | PMID: 23104886 | RRID:SCR_015899 | |
| Software, algorithm | edgeR | PMID: 19910308 | RRID:SCR_012802 | |
| Software, algorithm | DSS | PMID: 24561809 | RRID:SCR_002754 | |

## Cell derivation

Homozygous TauEGFP knock-in mice were bred with C57BL/6 mice (Jackson Labs) to generate heterozygous embryos (*Tucker et al., 2001*). The derivation of fibroblast cultures (TauEGF MEFs) was performed by isolating only the limbs from E13.5 embryos, which were then chopped into small pieces, trypsinized for 15 min at 37°C and plated in MEF media, containing 10% cosmic calf serum (Hyclone), 0.008% Beta mercaptoethanol (Sigma), MEM non-essential amino acids, Sodium Pyruvate, and Penicillin/Streptomycin (Pen/Strep) (all from Invitrogen). Neural progenitor cells (TauEGFP NPCs) were derived by harvesting the two cortical lobes from the brain of E12.5 embryos, which are then incubated in N3 media containing DMEM/F12, N2 supplement, Pen/Strep (all from Invitrogen) and 20 ug/ml Insulin (Sigma) at 37°C for 10 min. The cortical tissue was then gently dissociated with a 1 mL micro-pipette, passed through a 0.7 um filter before being plated in N3 media with EGF (20 ng/mL) and FGF (10 ng/mL) onto a cell culture dish that was previously coated with polyornithine (PO) (Sigma P3655) for at least 4 hr followed by laminin (LAM) (Sigma L2020) overnight.

For homozygous Dnmt3a/3b$^{fl/fl}$ MEFs, Dnmt3a$^{fl/fl}$ and Dnmt3b$^{fl/fl}$ mice on a C57BL/6 background were crossed to generate Dnmt3a/3b$^{fl/fl}$ mice (*Challen et al., 2014*; *Dodge et al., 2005*; *Kaneda et al., 2004*). Whole body MEFs were harvested from E13.5 embryos by removing the head, spinal cord and red organs, minced into small pieces, trypsinized and plated in MEF media on gelatinized flasks to derive fibroblast cultures (*Jozefczuk et al., 2012*). Both MEFs were expanded for three passages, while NPCs were expanded 3 to 4 passages prior to experiments.

### Direct conversion of fibroblast to neuron

Lentivirus was produced, and TauEGFP MEFs were co-infected with reverse tetracyclin transactivator (rtTA) and doxycycline (dox) inducible TetO-Ascl1 alone or with TetO-Brn2 and TetO-Mytl1 as previously described (*Marro and Yang, 2014*). Dox was added with fresh MEF media 16–20 hr post lentiviral infection, and cultures were switched to N3 +B27 media containing DMEM/F12, N2 and B27 supplements, Pen/Strep (all from Invitrogen) and 20 ug/ml Insulin with dox after 2 days. Cells were harvested at 48 hr, 5d and 22d post-dox induction for MethylC-seq. Control MEFs were not infected and harvested 48 hr after addition of dox. For 5d and 22d, cells were FAC (Fluorescence-activated cell)-sorted for TauEGFP +cells to select for cells that were reprogramming. To ensure that reprogramming efficiencies are comparable, immunofluorescence staining for Tuj1 was performed for each batch of cells at day 14, and only samples that average at least 20 Tuj1 +neurons per 10X field of view were used. In addition for FAC-sorted samples, only samples containing >5% TauEGFP +cells was used.

Dnmt3a/3b$^{fl/fl}$ MEFs were co-infected with rtTA, TetO-Ascl1 and either CreGFP or delCreGFP (both with constitutive FUW promoters). DelCreGFP contains a truncated and non-functional version of Cre and was used as a control. Fresh MEF media was added 16–20 hr after infection. Dox was then added between 2–4 days later to allow knockout of the Dnmt3a/b locus by Cre and degradation of the remaining protein before reprogramming with Ascl1.

### In-vitro differentiation of neural progenitor cells into neurons

P3-4 NPCs were seeded into cell culture plates that were previously coated with PO + LAM in N3 +EGF + FGF media. After one day, the media was replaced with fresh differentiation media containing N2 +B27 media, without EGF and FGF. Half media replacement with fresh N2 +B27 media was performed every alternate day. Control NPCs were harvested for MethylC-seq 2 days after seeding (without addition of differentiation media). Differentiating cells were harvested for MethylC-seq at 7d, 14d and 21d after addition of differentiation media and were FAC-sorted for TauEGFP +cells to select for neuronal cells.

### DNA extraction for MethylC-seq

DNA extraction was performed using the Qiagen DNeasy Blood and Tissue kit (#69506) with some modifications to the protocol. Fresh cells were washed once with PBS, then re-suspended in 900 uL ATL, 100 uL Proteinase K, 20 uL of RNAse A (Invitrogen 12091021) and 200 uL AL. Cell suspension is then incubated at 56°C for 10 min. Then 1 mL of 100% Ethanol is added and mixture was briefly vortexed before loading onto a spin-column. DNA is then washed on the column following Qiagen's protocol and finally eluted into 50–100 uL AE.

### Immunofluorescence and cell counting

Cells were fixed and counted as previously described (*Wapinski et al., 2017*). Briefly, cells were fixed with 4% paraformaldehyde at 14d post-dox, blocked with 5% cosmic calf serum (CCS), incubated with primary antibodies for at least an hour, followed by secondary antibodies for at least half an hour. Ten 10X images were then taken per biological replicate (MEFs derived from different embryos), and the relevant cells were counted. Tuj1 +neuronal cells were counted manually, and only cells with neurite extensions that are at least three times the length of the cell body diameter are included in the counts. ImageJ was used for counting DAPI, Dnmt3a + and Cre/delCreGfp + cells (*Schneider et al., 2012*). A threshold was first set to eliminate background noise and kept consistent for all images within a single biological replicate. Then 'Watershed' was ran to distinguish distinct nuclei, and 'Analyze particles' was used (size = 0.01 Infinity) to count the number of cells. For each replicate, the number of cells per 10X view was taken as an average of all the images taken.

### Antibodies

Rabbit anti-Tubb3 (Tuj1, Covance MRB-435P), rabbit anti-Dnmt3a (H-295, Santa Cruz Biotech sc-20703), chicken anti-GFP (Abcam ab13970), rabbit anti-HSP90 (cell signaling 4874), rabbit anti-H3 (cell signaling 9712). Secondary Alexa-conjugated antibodies were used at 1:1000 (all from Invitrogen).

## MethylC-seq

MethylC-seq libraries were prepared as previously described (*Urich et al., 2015*), except regular lambda DNA (Promega cat. # D1501) isolated from *dcm* +E.Coli was spiked into samples as the control for non-conversion rate. MethylC-seq libraries were sequenced on Illumina HiSeq 2500. Non-conversion rate was computed from each sample after excluding CAG and CTG trinucleotides from lambda DNA sequence. MethylC-seq reads were mapped to mm9 reference genome using MethylPy (https://bitbucket.org/schultzmattd/methylpy/wiki/Home) (*Lister et al., 2013*; *Ma et al., 2014*). To ensure the correct calling of mCG and mCH, we removed cytosine positions with potential single nucleotide variants located at cytosines or immediately downstream of cytosines as previously described (*Luo et al., 2016*).

To compare the methylome of iN cells to primary neurons and other mouse tissues, methylome reads of NeuN +neurons (GSM1173786), NeuN- glia (GSM1173787), Camk2a + excitatory neurons (GSM1541958 and GSM1541959), cortex (GSM1051153), cerebellum (GSM1051151), olfactory bulb (GSM1051159), heart (GSM1051154), kidney (GSM1051156), lung (GSM1051158) and pancreas (GSM1051160) were downloaded from NCBI SRA and processed identically as iN cell methylome data.

Differentially methylated sites were identified between pairs of iN cell samples using DSS with FDR < 0.1 (*Feng et al., 2014*). Ascl1 DMSs were identified by pairwise comparison between MEF and all Ascl1 iN cell samples. All pairwise DMSs showing reduced mCG during reprogramming were combined and assigned into categories based on the kinetics of mCG reduction. Specifically, the dynamics of each Ascl1 DMS was represented by a vector containing three elements with each element indicating the transition steps from MEF to Ascl1 2d, Ascl1 2d to Ascl1 5d and Ascl1 5d to Ascl1 22d, respectively. The sign of each elements indicates the direction of change. DMSs between adjacent time points (e.g. Ascl1 5d vs. 22d) were represented by a transition step with value −1. DMSs between nonadjacent time points were represented by −0.5 for DMSs called between samples with two transition step distance (e.g. Ascl1 2d vs. 22d) or −0.33 for DMSs called between samples with three transition step distance (e.g. MEF vs. 22d). Examples of DMS dynamics are shown below:

| Pairwise DMS calls (mCG change) | MEF to 2d | 2d to 5d | 5d to 22d |
|---|---|---|---|
| MEF > 2d | −1 | 0 | 0 |
| 5d > 22d | 0 | 0 | −1 |
| 2d > 22d | 0 | −0.5 | −0.5 |
| MEF > 22d | −0.33 | −0.33 | −0.33 |
| MEF > 5d & MEF > 22d & 5d > 22d | −0.5 | −0.5 | −1 |
| 2d > 5d & 2d > 22d & 5d > 22d | 0 | −1 | −1 |

BAM DMSs were similarly identified, combined and grouped between MEF and all BAM iN cell samples.

## ChIP-seq data

Ascl1 and Brn2 ChIP-seq dataset for iN cell samples were previously published in GSE43916 (*Wapinski et al., 2013*) - Ascl1 MEF (SRR935631, SRR935632), Ascl1 BAM (SRR935633, SRR935635, SRR935634), Brn2 MEF (SRR935643, SRR935644), Brn2 BAM (SRR935645, SRR935646), Brn2 NSC (SRR935647, SRR935648). H3K27me3 ChIP-seq peaks for adult mouse cerebellum and P0 mouse hindbrain were downloaded from ENCODE accession ENCFF112CIF and ENCFF953WTE, respectively, and converted to mm9 coordinates. Sequencing reads were mapped to mm9 reference genome using bowtie2 2.1.0 (*Langmead and Salzberg, 2012*). ChIP-seq peaks were identified using MACS2 2.0.10 with q value < 0.01. Peaks called from the two replicates of Brn2 BAM ChIP-seq were combined. To analyze DNA demethylation induced by TF bindings, Ascl1 and Brn2 peaks that overlapped with lowly methylate regions (UMRs or LMRs) identified from Ascl1 2d methylome were removed. UMRs and LMRs were identified using MethylSeekR with m = 0.5% and 5% FDR (*Burger et al., 2013*).

### RNA-seq data

For all reprocessing of published RNA-seq data, RNA-seq reads were downloaded from NCBI SRA and mapped to mm9 Refseq gene annotation using STAR 2.4.0 (*Dobin et al., 2013*). Mapped RNA-seq reads were counted and summarized to the gene level with HTSeq 0.6.1 followed by normalization and computation of RPKM using edgeR 3.8.6 (*Anders et al., 2015*; *Robinson et al., 2010*). Differentially expressed genes were identified using edgeR 3.8.6 with FDR < 0.05 and Fold Change >= 2 (*Robinson et al., 2010*).

RNA-seq dataset of iN cell reprogramming was downloaded from GSE43916 (*Wapinski et al., 2013*). Single cell gene expression dataset of iN cell reprogramming was previously published (*Treutlein et al., 2016*). Mouse tissue RNA-seq dataset was downloaded from NCBI GEO accessions cerebellum (GSM723768), cortex (GSM723769), heart (GSM723770), kidney (GSM723771), lung (GSM723773) and olfactory bulb (GSM850911) (*Shen et al., 2012*). Cerebellum developmental RNA-seq dataset was downloaded from NCBI SRA accessions P7 (SRR1557065, SRR1557066, SRR1557067), P14 (SRR1557068, SRR1557069, SRR1557070) and P60 (SRR1557071, SRR1557072, SRR1557073). Dnmt3a -/-;Nes-cre gene expression dataset was downloaded from NCBI GEO accessions WT (GSM1464557, GSM1464558, GSM1464559) and Dnmt3a KO (GSM1464560, GSM1464561, GSM1464562).

### TF binding motif enrichment analysis

TF binding position weight matrices (PWM) were obtained from the MEME motif database and scanned across the mouse mm9 reference genome to identify hits using FIMO (–output-pthresh 1E-5, – max-stored-scores 500000 and –max-strand) (*Bailey et al., 2009*; *Grant et al., 2011*). DMSs were extended 250 bp both upstream and downstream for overlapping with TF binding motif hits. The overlap between TF binding motif hits and DMSs (extended ±250 bp) were determined requiring a minimal of 1 bp overlap. The enrichment of TF binding motifs in DMSs was assessed using DMRs (extended 250 bp from center) identified across adult mouse tissues (tissue DMRs) as the background (*Hon et al., 2013*). The overlaps between TF binding motif hits and iN cell DMS categories (foreground) was compared to the overlaps between TF binding motifs hits and tissue DMRs (background) using the hypergeometric test (MATLAB hygecdf). A TF binding motif is considered significantly enriched or depleted if the hypergeometric test resulted in q value < 1E-5.

### Accession numbers

MethylC-seq data files were deposited to NCBI GEO accession GSE111283. Methylome profiles are visualized using an AnnoJ browser at http://neomorph.salk.edu/iN_transdifferentiation.php.

## Acknowledgements

The work is supported by NIH P50-HG007735 (HYC), CIRM RB5-07466 (HYC, MW) and NIH R01 DK092883 (MAG). JRE and HYC are Howard Hughes Medical Institute investigators.

## Additional information

### Funding

| Funder | Author |
| --- | --- |
| National Institute of Diabetes and Digestive and Kidney Diseases | Margaret Goodell |
| Howard Hughes Medical Institute | Joseph R Ecker<br>Howard Y Chang |
| National Human Genome Research Institute | Howard Y Chang |
| California Institute for Regenerative Medicine | Howard Y Chang<br>Marius Wernig |

The funders had no role in study design, data collection and interpretation, or the decision to submit the work for publication.

## Author contributions
Chongyuan Luo, Qian Yi Lee, Formal analysis, Investigation, Visualization, Methodology, Writing—original draft, Writing—review and editing; Orly Wapinski, Investigation, Writing—original draft, Writing—review and editing; Rosa Castanon, Joseph R Nery, Moritz Mall, Michael S Kareta, Sean M Cullen, Investigation, Methodology, Writing—review and editing; Margaret A Goodell, Howard Y Chang, Marius Wernig, Joseph R Ecker, Funding acquisition, Investigation, Methodology, Project administration, Writing—review and editing

## Author ORCIDs
Chongyuan Luo http://orcid.org/0000-0002-8541-0695
Margaret A Goodell http://orcid.org/0000-0003-1111-2932
Howard Y Chang http://orcid.org/0000-0002-9459-4393
Joseph R Ecker http://orcid.org/0000-0001-5799-5895

## Decision letter and Author response
Decision letter https://doi.org/10.7554/eLife.40197.019
Author response https://doi.org/10.7554/eLife.40197.020

# Additional files

## Supplementary files
• Supplementary file 1. Summary of MethylC-seq experiments.
DOI: https://doi.org/10.7554/eLife.40197.013
• Supplementary file 2. Enriched gene ontology terms in gene clusters defined by gene body mCH.
DOI: https://doi.org/10.7554/eLife.40197.014
• Supplementary file 3. Differentially methylated sites (DMSs) identified during direct reprogramming driven by Ascl1-alone or BAM.
DOI: https://doi.org/10.7554/eLife.40197.015
• Transparent reporting form
DOI: https://doi.org/10.7554/eLife.40197.016

## Data availability
MethylC-seq data files were deposited to NCBI GEO accession GSE111283. Methylome profiles are visualized using an AnnoJ browser at http://neomorph.salk.edu/iN_transdifferentiation.php.

The following dataset was generated:

| Author(s) | Year | Dataset title | Dataset URL | Database and Identifier |
|---|---|---|---|---|
| Luo C, Lee QY, Wapinski OL, Castanon R, Nery JR, Cullen SM, Goodell MA, Chang HY, Wernig M, Ecker JR | 2018 | Global DNA methylation remodeling during direct reprogramming from fibroblast to neuron | https://www.ncbi.nlm.nih.gov/geo/query/acc.cgi?acc=GSE111283 | NCBI Gene Expression Omnibus, GSE111283 |

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
