## [Decision Letter]

Thank you for submitting your article "Global DNA methylation remodeling during direct reprogramming of fibroblasts to neurons" for consideration by *eLife*. Your article has been reviewed by two peer reviewers, and the evaluation has been overseen by a Reviewing and Jessica Tyler as the Senior Editor. The reviewers have opted to remain anonymous.

The reviewers have discussed the reviews with one another and the Reviewing Editor has drafted this decision to help you prepare a revised submission.

Both reviewers felt that the work was exciting and important. The major contribution of this work is a detailed description of the DNA methylation status upon reprogramming fibroblasts to neurons as well as a convincing demonstration of how DNA methylation levels in reprogrammed cells mostly reflect the methylation status of mature neurons in a mouse brain, and they discern the critical transcription factors involved. There are no major outstanding concerns regarding the experiments or the conclusions of the work. There are a few minor concerns that could be addressed:

Minor Comments:

1) Figure 2G demonstrates a down-regulation of developmentally up-regulated genes in Dnmt3a KO cerebellum. Can the authors speculate on why the ablation of a de novo DNA methylase leads to down-regulation of normally up-regulated genes – i.e. indirect effect?

2) In regards to Figure 4 the authors speculate that ASCL1 can recruit demethylation machinery as it is frequently found at DMSs that lose methylation early in Ascl1-only reprogramming. Could the authors indicate whether this loss may also occur through passive dilution/lack of recruitment of the maintenance methylase?

3) Figure 5B shows Ascl1 at the promoter of Dnmt3a by ChIP – is there any data that Ascl1 binding induces Dnmt3a expression? Knockdown/knockout of Ascl1 affecting Dnmt3a expression levels for example?

4) In Figure 5C the large population of Dnmt3a positive cells that remain despite Dox treatment seems to suggest these cells may be selected for in the process of reprogramming. The Cre recombinase strategy is certainly not 100% efficient, but seems to work very well in this system according to Figure 5D. Could the authors provide some explanation for why this population is "a large fraction"?

5) The authors state "We identified a large number (2,098, Cluster 15 and 16) of genes that were clearly hypermethylated in MEFs, and immature iN cells but were strongly depleted of mCH in BAM 22d iN cells (box in Figure 1C)". I suggest a revision of the statement here because, according to the key for normalized mCH values, hypermethylation corresponds to red and genes in cluster 15 and 16 in MEFs show instead medium levels of methylation (pale blue).

In the text: "Through correlating with transcriptome analysis of iN cell reprogramming, we found that genes in Cluster 15 and 16 were actively expressed in successfully reprogrammed iN cells (Figure 1E and Figure 1—figure supplement 1E) (Treutlein et al., 2016)" Figure —figure supplement 1E is missing.

6) The authors ranked genes by gene body mCH levels in BAM 5d. The authors state "Notably, genes with early mCH accumulation showed strongly reduced expression in the late stage of neuronal reprogramming (BAM 22d, Figure 2B-C)." I suggest a careful consideration with this statement since, looking at Figure 2A-C, one could also say that early accumulation of mCH can predict later transcriptional expression/up-regulation. For example, looking at BAM 13d, early mCH accumulation (BAM 5d) would correlate first with gene activation at BAM 13d, then with strongly reduced expression at BAM 22d. This type of correlative analysis would be more appropriate having gene expression data from BAM 5d and BAM 22d and even better by performing RNA/DNA parallel purification from the same sample and then generating the data sets. Same for where the authors state in the figure legend to Figure 2 "Early mCH accumulation is strongly correlated to later gene repression".

7) The authors state "Surprisingly, we found different patterns depending on the fold-change of gene expression: Mildly up-regulated genes accumulated intermediate mCH levels between down-regulated and static genes whereas strongly up-regulated genes and static genes were close to the MEF baseline of mCH levels (Figure 2D, E). These results suggest a model that mCH is preferentially targeted to mildly dynamic genes and modulates their expression during reprogramming, for strongly differentially regulated genes, however, mCH is mostly accumulating at down-regulated genes." I suggest to clarify with more information the reference points used to determine differential gene expression to create the categories Down, Up, Static. It is not clear if the differential gene expression is between BAM 2d and BAM22d. Considering the data from Gabel et al., 2015 normalizing to gene size in the analysis of Figure 2D, E may be relevant since a longer gene could accumulate more mCH in association with gene repression.

8) In the sentence "To explore the interaction between CG methylation and TF binding, in particular the binding of ASCL1, BRN2 and MYT1L, we identified 10,075 and 15,093 differentially methylated sites (DMSs) showing reduced mCG during Ascl1-only and BAM induced reprogramming, respectively (Figure 4A, C and Supplementary file 3)." 1) The numbers of DMSs 10,075 and 15,093, don't match the numbers that are summed up from the groups in the Figure 4A, C. 2) In the figure legend the information about which pair of iN cells were compared in the determination of DMSs is missing. For Ascl1-only, is it 22d vs. 2d? And for BAM-only, 22d vs. 5d? Finally, for those exact locations, were mCG estimations for 5d and MEF plotted? 3) "only 0.6% (99/15,093) BAM DMSs were overlapped with ASCL1 peaks (Figure 4D)". I suggest to verify if the percentage indicated here is correct.

9) In the sentence "Among the sites that become demethylated in Ascl1-only reprogramming we found a striking co-enrichment of ASCL1 binding, in particular at early demethylating sites (Figure 4B and Figure 4—figure supplement 1C)". The ChIP-seq data for Ascl1 in MEFs show Ascl1 binding in MEFs preceding direct reprogramming, which is surprising. Same as the ChIP-seq data for BRN2 that show binding in MEFs preceding direct reprogramming. I suggest to add a brief sentence to justify this observation based on prior data, so to clarify the result.

10) "we found only 0.3% (38/15,093) BAM DMSs overlapped with MYT1L binding sites" I suggest to verify if the percentage indicated here is correct. The DMSs reported in the Supplementary file 3A, B are specific CG sites 1bp. It would help to understand the data if the authors could elaborate on how the overlapping analysis was done: given the information provided, it seems that between the genomic coordinates of ChIP-seq peaks and the genomic coordinates of a specific CG site differentially methylated. Same for the motif enrichment analysis: I would add a brief sentence explaining how it was done because in the text the authors mention regions "To further illuminate the nature of the demethylating regions" Are these UMR/LMRs?

11) The authors state "Our results collectively suggest that mCH facilitates neuronal maturation during development through gene expression". Is it through gene expression or gene repression here?

12) Optional: In Figure 1B it would be good to include the cortex MethylC-seq data from Hon et al., 2013 in order to examine if BAM 22d iN cells also cluster closely to cortex tissue.

The full reviews are given here for your interest:

*Reviewer #1:*

This work seeks to identify the genome-wide changes in DNA methylation that occur during the process of direct reprogramming of fibroblasts to induced neurons. In doing so the group examines two reprogramming strategies that are composed of either expressing a neuron-inducing transcription factor (Ascl1) or expressing the inducing transcription factor in combination with two other transcription factors that repress a myogenic state. They compare the DNA methylation status of these two strategies not only to each other but also to mature in vivo neurons in order to demonstrate the capacity of reprogramming to faithfully exhibit molecular markers of the intended cell types. These comparisons allow for an in vitro examination of the role that these transcription factors may also have in vivo neuronal development. Additionally, on a mechanistic level the authors indicate that the progressive increase in DNA methylation is due in large part to the increased expression of one of the de novo methylases – Dnmt3a.

The major contribution of this work is a detailed description of the DNA methylation status in induced neurons as well as a convincing demonstration of how DNA methylation levels in reprogrammed cells mostly reflect the methylation status of mature neurons in a mouse brain. This has not been sufficiently demonstrated previously. The authors demonstrate how a cellular reprogramming system may indicate which epigenetic marks are important to in vivo development and differentiation such as mCH DNA methylation in neuronal development in this case. Furthermore, the authors show that reprogramming with the three transcription factors (BAM) more accurately recapitulates the gene expression and methylation levels of mature neurons as opposed to reprogramming with Ascl1 alone indicating that reprogramming is aided by the repression of alternative states via epigenetic regulation.

There are no major outstanding concerns regarding the experiments or the conclusions of the work. There are a few minor concerns that could be addressed:

1) Figure 2G demonstrates a down-regulation of developmentally up-regulated genes in Dnmt3a KO cerebellum. Can the authors speculate on why the ablation of a de novo DNA methylase leads to down-regulation of normally up-regulated genes – i.e. indirect effect?

2) In regards to Figure 4 the authors speculate that ASCL1 can recruit demethylation machinery as it is frequently found at DMSs that lose methylation early in Ascl1-only reprogramming. Could the authors indicate whether this loss may also occur through passive dilution/lack of recruitment of the maintenance methylase?

3) Figure 5B shows Ascl1 at the promoter of Dnmt3a by ChIP – is there any data that Ascl1 binding induces Dnmt3a expression? Knockdown/knockout of Ascl1 affecting Dnmt3a expression levels for example?

4) In Figure 5C the large population of Dnmt3a positive cells that remain despite Dox treatment seems to suggest these cells may be selected for in the process of reprogramming. The Cre recombinase strategy is certainly not 100% efficient, but seems to work very well in this system according to Figure 5D. Could the authors provide some explanation for why this population is "a large fraction"?

*Reviewer #2:*

In the current manuscript entitled "Global DNA methylation remodelling during direct reprogramming of fibroblasts to neurons" Luo & Lee et al., generate whole-genome base-resolution methylomes by MethylC-seq to characterize the global reconfiguration of the DNA methylome through the induction of a neuronal phenotype (iN) in mouse embryonic fibroblast using a lineage instructive approach, ectopically expressing the pioneer transcription factor Ascl1 alone or in combination with Brn2 and Mytl1. They show that the two reprograming strategies led to considerably different DNA methylation remodelling, with BAM factors producing a pattern that is more comparable to mature cortical neurons than Ascl1 alone. These factors are required, yet not sufficient, to acquire the DNA methylome of mature cortical neurons observed during in vivo neuron differentiation, acknowledging consistencies and differences between iN cells and mature neurons.

The authors have strong expertise in the fields of trans differentiation, particularly with the induction of fibroblasts to neurons by employing a defined set of transcription factors, such as Ascl1, Brn2 and Mytl1, whose ectopic expression in fibroblasts has been previously shown to induce a functional neuronal phenotype by these and other groups.

An exciting part of the work is the interrogation of the contribution of these TF to the reconfiguration of the DNA methylome. The authors show the accumulation of mCH in iN cells, an epigenomic signature of neuronal differentiation and of mature cortical neurons, in association with gene down-regulation and also examine mCG dynamics during conversion, an aspect not examined until this work on iN cells.

Interestingly, the acquisition of mCG associates with the suppression of fibroblast and myogenic programs and conditional ablation of Dnmt3a reduces the efficiency of iN cells generation from fibroblast, indicating that de novo methylation is required for efficient direct conversion to iN cells.

The paper is well-written, the methods fit with the aims of the study with a clearly described approach, the results are well-articulated and the conclusions are appropriate, given the results. I am overall enthusiastic about the work presented by the authors as no pre-existing study has been published addressing the reprograming of the DNA methylome upon direct conversion of fibroblasts to neurons. There are minor concerns and details missing about the data analysis not entirely clear that the authors ought to address. Despite these concerns, I believe that this manuscript meets the publication criteria for *ELife*.

The authors state "We identified a large number (2,098, Cluster 15 and 16) of genes that were clearly hypermethylated in MEFs, and immature iN cells but were strongly depleted of mCH in BAM 22d iN cells (box in Figure 1C)". I suggest a revision of the statement here because, according to the key for normalized mCH values, hypermethylation corresponds to red and genes in cluster 15 and 16 in MEFs show instead medium levels of methylation (pale blue).

In the text: "Through correlating with transcriptome analysis of iN cell reprogramming, we found that genes in Cluster 15 and 16 were actively expressed in successfully reprogrammed iN cells (Figure 1E and Figure 1—figure supplement 1E) (Treutlein et al., 2016)" Figure 1—figure supplement 1E is missing.

The authors ranked genes by gene body mCH levels in BAM 5d. The authors state "Notably, genes with early mCH accumulation showed strongly reduced expression in the late stage of neuronal reprogramming (BAM 22d, Figure 2B-C)." I suggest a careful consideration with this statement since, looking at Figure 2a-c, one could also say that early accumulation of mCH can predict later transcriptional expression/up-regulation. For example, looking at BAM 13d, early mCH accumulation (BAM 5d) would correlate first with gene activation at BAM 13d, then with strongly reduced expression at BAM 22d. This type of correlative analysis would be more appropriate having gene expression data from BAM 5d and BAM 22d and even better by performing RNA/DNA parallel purification from the same sample and then generating the data sets. Same for where the authors state in the figure legend to Figure 2 "Early mCH accumulation is strongly correlated to later gene repression".

The authors state "Surprisingly, we found different patterns depending on the fold-change of gene expression: Mildly up-regulated genes accumulated intermediate mCH levels between down-regulated and static genes whereas strongly up-regulated genes and static genes were close to the MEF baseline of mCH levels (Figure 2D, E). These results suggest a model that mCH is preferentially targeted to mildly dynamic genes and modulates their expression during reprogramming, for strongly differentially regulated genes, however, mCH is mostly accumulating at down-regulated genes." I suggest to clarify with more information the reference points used to determine differential gene expression to create the categories Down, Up, Static. It is not clear if the differential gene expression is between BAM 2d and BAM22d. Considering the data from Gabel et al., 2015 normalizing to gene size in the analysis of Figure 2D, E may be relevant since a longer gene could accumulate more mCH in association with gene repression.

In the sentence "To explore the interaction between CG methylation and TF binding, in particular the binding of ASCL1, BRN2 and MYT1L, we identified 10,075 and 15,093 differentially methylated sites (DMSs) showing reduced mCG during Ascl1-only and BAM induced reprogramming, respectively (Figure 4A, C and Supplementary file 3).” 1) The numbers of DMSs 10,075 and 15,093, don't match the numbers that are summed up from the groups in the Figure 4A, C. 2) In the figure legend the information about which pair of iN cells were compared in the determination of DMSs is missing. For Ascl1-only, is it 22d vs. 2d? And for BAM-only, 22d vs. 5d? Finally, for those exact locations, were mCG estimations for 5d and MEF plotted? 3) "only 0.6% (99/15,093) BAM DMSs were overlapped with ASCL1 peaks (Figure 4D)" I suggest to verify if the percentage indicated here is correct.

In the sentence "Among the sites that become demethylated in Ascl1-only reprogramming we found a striking co-enrichment of ASCL1 binding, in particular at early demethylating sites (Figure 4B and Figure 4—figure supplement 1C)". The ChIP-seq data for Ascl1 in MEFs show Ascl1 binding in MEFs preceding direct reprogramming, which is surprising. Same as the ChIP-seq data for BRN2 that show binding in MEFs preceding direct reprogramming. I suggest to add a brief sentence to justify this observation based on prior data, so to clarify the result.

"we found only 0.3% (38/15,093) BAM DMSs overlapped with MYT1L binding sites" I suggest to verify if the percentage indicated here is correct. The DMSs reported in the Supplementary file 3A, B are specific CG sites 1bp. It would help to understand the data if the authors could elaborate on how the overlapping analysis was done: given the information provided, it seems that between the genomic coordinates of ChIP-seq peaks and the genomic coordinates of a specific CG site differentially methylated. Same for the motif enrichment analysis: I would add a brief sentence explaining how it was done because in the text the authors mention regions "To further illuminate the nature of the demethylating regions" Are these UMR/LMRs?

The authors state "Our results collectively suggest that mCH facilitates neuronal maturation during development through gene expression". Is it through gene expression or gene repression here?

Optional: In Figure 1B it would be good to include the cortex MethylC-seq data from Hon et al., 2013 in order to examine if BAM 22d iN cells also cluster closely to cortex tissue.

---

## [Author Response]

Minor Comments:*1) Figure 2G demonstrates a down-regulation of developmentally up-regulated genes in Dnmt3a KO cerebellum. Can the authors speculate on why the ablation of a* de novo *DNA methylase leads to down-regulation of normally up-regulated genes – i.e. indirect effect?*

We agree with the reviewer that it is intriguing that the ablation of Dnmt3a leads to downregulation of developmentally upregulated genes. Given the role of mC as a repressive epigenetic mark, it seems to be paradoxical that the major de novo DNA methyltransferase DNMT3A is required for supporting gene expression. We considered two hypotheses that may explain why mC mediated by DNMT3A can contribute positively to transcription.

1) DNMT3A mediated mC antagonizes polycomb repression at certain loci and is associated with active gene expression.

To test this hypothesis, we first asked whether developmentally upregulated genes are more likely to be targeted by polycomb repressive complex, using H3K27me3 as a marker for polycomb regulation. However, we found that a comparable fraction (17%, p-value = 0.87, fisher-exact test) of developmentally up- and down- regulated genes were regulated by polycomb complex.

Secondly, we asked whether developmentally upregulated genes associated with H3K27me3 peaks are more strongly down-regulated upon the ablation of Dnmt3a, compared to genes that do not overlap with H3K27me3. However, for either developmentally up- or down- regulated genes, genes enriched in H3K27me3 do not show significantly different mis-regulation compared to genes without H3K27me3 enrichment.

Together, our results suggest that the observed positive correlation between mC and gene expression was not majorly contributed by the antagonism between mC and polycomb repression. This analysis is included in the revised manuscript as Figure 2—figure supplement 1C.

This analysis have been incorporated into the revised manuscript.

“Unexpectedly, Dnmt3a ablation also led to a downregulation of developmentally upregulated genes (p = 6x10^-15^, Figure 2G), suggesting that mC support gene expression at these loci. […] Therefore, these results do not support the model that mC indirectly promote gene expression by antagonizing polycomb repression.”

2) DNMT3A indirectly support gene expression by contributing to the accumulation of hydroxymethylcytosine in CG dinucleotide context (hmCG).

Cerebellar neurons accumulate a high level of hmCG, which is associated with active gene expression. It was recently shown that hmCG cannot interact with MECP2 (Gabel et al., 2015; Mellén et al., 2017). Thus the oxidation of mCG abolishes its repressive function. We speculate that DNMT3A, through catalyzing de novo mCG, provides the necessary substrate for TET methylcytosine dioxygenase to produce hmCG. To ask whether the accumulation of hmCG is involved in gene activation during development, we compared different cytosine modifications between developmentally up- and down- genes using an oxBS-seq dataset generated from cerebellar granule cells (Mellén et al., 2017). Interestingly, developmentally up-regulated genes showed significantly higher level of gene body hmCG, but not other types of cytosine methylation, than developmentally down-regulated genes (p = 1.3 x 10^-6^, Wilcoxon rank sum test) and static genes (p = 2.4 x 10^-55^, Wilcoxon rank sum test). Thus, DNMT3A could indirectly support gene expression by mechanistically contribute to gene body hmCG.

This analysis have been incorporated into the revised manuscript.

“We considered a second hypothesis that Dnmt3a indirectly supports developmental gene activation by providing the substrate for hydroxymethylcytosine in a CG dinucleotide context (hmCG). […] The result suggests that the activation of developmentally upregulated genes is associated with hmCG accumulation, which may indirectly require DNMT3A for contributing the substrate (mCG) for TET methylcytosine dioxygenases to produce hmCG.”

2) In regards to Figure 4 the authors speculate that ASCL1 can recruit demethylation machinery as it is frequently found at DMSs that lose methylation early in Ascl1-only reprogramming. Could the authors indicate whether this loss may also occur through passive dilution/lack of recruitment of the maintenance methylase?

We have previously shown that cells exit the cell cycle within 2 days of Ascl1 induction based on significant reduction in Ki67 positive cells (Davila et al., 2013) and homogenous downregulation of cell cycle genes from single cell RNA-seq data (Treutlein et al., 2016). Thus, the observed reduction of mCG at ASCL1 binding site is likely mediated by direct demethylation instead of passive dilution. We have included this discussion in the revised manuscript.

“The reduction of local mCG during the reprogramming of fibroblasts to iN cells is likely mediated by direct demethylation instead of passive dilution, since we have previously shown that cells exit cell cycle within 2 days of Ascl1 induction based on significant reduction in Ki67 positive cells (Davila et al., 2013) and homogenous downregulation of cell cycle genes from single cell RNA-seq data (Treutlein et al., 2016).”

3) Figure 5B shows Ascl1 at the promoter of Dnmt3a by ChIP – is there any data that Ascl1 binding induces Dnmt3a expression? Knockdown/knockout of Ascl1 affecting Dnmt3a expression levels for example?

We agree with the reviewer that the original manuscript did not present evidence to support the functional regulation of Dnmt3a by ASCL1 binding. After examining published data from both mouse and human that are obtained from knock-down or over-expression of Ascl1, we cannot conclusively determine whether Ascl1 functionally regulates Dnmt3a expression.

1) Dnmt3a expression is not significantly different between MEF and MEF cells over-expressing Ascl1 for 2 days (fold change = 1.38, p-value = 0.28, t-test).

2) ASCL1 down regulation in human U87MG glioma cell line does not induce Dnmt3a mis-regulation (fold change = 1.07, p-value = 0.25, t-test, GSE76652) (Donakonda et al., 2017).

3) Ascl1 knock-out in human glioblastoma stem cells does not alter Dnmt3a expression (fold change = 1.13, p-value = 0.1, t-test, GSE87615) (Park et al., 2017).

4) Ascl1 over-expression in Ascl1 knock-out human glioblastoma stem cell line leads to down-regulation of Dnmt3a (fold change = 0.81, p-value = 0.006, GSE87617) (Park et al., 2017).

We have revised the manuscript to clearly indicate that this remains an open question.

“However, Ascl1 over-expression is insufficient to activate Dnmt3a expression since Dnmt3a was not significantly upregulated 2 days after Ascl1 induction. It remains unclear whether the binding of ASCL1 to the Dnmt3a promoter functionally contribute to Dnmt3a activation during reprogramming.”

4) In Figure 5C the large population of Dnmt3a positive cells that remain despite Dox treatment seems to suggest these cells may be selected for in the process of reprogramming. The Cre recombinase strategy is certainly not 100% efficient, but seems to work very well in this system according to Figure 5D. Could the authors provide some explanation for why this population is "a large fraction"?

We thank the reviewer for this comment. Figures 5 and Figure 5—figure supplement 1E show that in the Cre infected population, we obtained a 75% efficiency in knocking out Dnmt3a, even though based on Cre staining, we see that infection efficiency is over 90% (data not shown). We also showed in Figure 5—figure supplement 1B that many of the cells expressing Cre also express Dnmt3a, suggesting either incomplete degradation of Dnmt3a or escape from Cre recombination. However, we agree that saying this population is “a large fraction” is inaccurate, and we have modified the manuscript as below:

“Despite efficient Dnmt3a depletion in the total fibroblast population (Figure 5D, Figure 5—figure supplement 1C) we observed a number of Dnmt3a-positive iN cells in the Cre-infected group, presumably due to incomplete degradation of Dnmt3a or escape from Cre recombination (Figure 5C, Figure 5—figure supplement 1D-F).”

5) The authors state "We identified a large number (2,098, Cluster 15 and 16) of genes that were clearly hypermethylated in MEFs, and immature iN cells but were strongly depleted of mCH in BAM 22d iN cells (box in Figure 1C)". I suggest a revision of the statement here because, according to the key for normalized mCH values, hypermethylation corresponds to red and genes in cluster 15 and 16 in MEFs show instead medium levels of methylation (pale blue).

We agree with the reviewer that in immature iN cells and Ascl1 22d iN cells, the mCH level of clusters 15 and 16 can be more appropriately described as medium. The manuscript has been revised as below:

“We identified a large number (2,098, Cluster 15 and 16) of genes that were strongly depleted of mCH in BAM 22d iN cells but show medium levels of mCH in immature iN cells and Ascl1 22d iN cells (box in Figure 1C).”

In the text: "Through correlating with transcriptome analysis of iN cell reprogramming, we found that genes in Cluster 15 and 16 were actively expressed in successfully reprogrammed iN cells (Figure 1E and Figure 1—figure supplement 1E) (Treutlein et al., 2016)" Figure —figure supplement 1E is missing.

We thank the reviewer for noticing this oversight. Figure 1—figure supplement 1D instead of Figure 1—figure supplement 1E should be referenced here. This error has been corrected in the revised manuscript.

6) The authors ranked genes by gene body mCH levels in BAM 5d. The authors state "Notably, genes with early mCH accumulation showed strongly reduced expression in the late stage of neuronal reprogramming (BAM 22d, Figure 2B-C)." I suggest a careful consideration with this statement since, looking at Figure 2A-C, one could also say that early accumulation of mCH can predict later transcriptional expression/up-regulation. For example, looking at BAM 13d, early mCH accumulation (BAM 5d) would correlate first with gene activation at BAM 13d, then with strongly reduced expression at BAM 22d. This type of correlative analysis would be more appropriate having gene expression data from BAM 5d and BAM 22d and even better by performing RNA/DNA parallel purification from the same sample and then generating the data sets. Same for where the authors state in the figure legend to Figure 2 "Early mCH accumulation is strongly correlated to later gene repression".

As suggested by the reviewer, we performed statistical analyses to ask whether genes showing early mCH accumulation are enriched in genes upregulated at BAM 13d time point. Indeed, comparing to MEF, both up-regulated and down-regulated gene in BAM 13d are moderately enriched in genes showing early mCH accumulation (Figure 2C). We thus concluded that early mCH accumulation is correlated with genes showing dynamic expression during reprogramming, and most strikingly repressed genes in matured iN cells (BAM 22d). The manuscript has been revised as below:

“Genes showing early mCH accumulation were strongly enriched in downregulated genes (compared to MEF) in BAM 22d iN cells, and to a less extent enriched in both upregulated and downregulated genes in BAM 13d iN cells (Figure 2B and C). Thus early mCH accumulation is correlated with genes showing dynamic expression during reprogramming, and most strikingly with genes repressed in matured iN cells (BAM 22d).”

7) The authors state "Surprisingly, we found different patterns depending on the fold-change of gene expression: Mildly up-regulated genes accumulated intermediate mCH levels between down-regulated and static genes whereas strongly up-regulated genes and static genes were close to the MEF baseline of mCH levels (Figure 2D, E). These results suggest a model that mCH is preferentially targeted to mildly dynamic genes and modulates their expression during reprogramming, for strongly differentially regulated genes, however, mCH is mostly accumulating at down-regulated genes." I suggest to clarify with more information the reference points used to determine differential gene expression to create the categories Down, Up, Static. It is not clear if the differential gene expression is between BAM 2d and BAM22d.

It appears that due to an error during the assembly of manuscript that Figure 2D-E were misinterpreted in the original manuscript. In Figure 2D-E and Figure 2—figure supplement 1, genes were stratified by average expression level across all time points of reprogramming, instead of dynamic expression patterns. We apologize for this error and we have revised the text related to this result:

“Surprisingly, we found different patterns depending on gene expression levels: lowly expressed genes accumulated high levels of mCH regardless of their developmental dynamics (Figure 2D; Figure 2—figure supplement 1A), whereas for actively expressed genes, gain of mCH is specific to developmentally downregulated genes; the mCH levels of upregulated and static genes were close to the MEF baseline (Figure 2E and Figure 2—figure supplement 1B). These results suggest a model that mCH is preferentially targeted to two main gene groups – constitutively repressed genes and actively expressed genes showing developmental downregulation.”

Considering the data from Gabel et al., 2015 normalizing to gene size in the analysis of Figure 2D, E may be relevant since a longer gene could accumulate more mCH in association with gene repression.

The reviewer has raised an interesting point that gene body mCH accumulation may be related to gene length as described in Gabel et al., 2015. In the revised manuscript, we have compared the relation between gene length and gene body mCH level in both mouse cortex and iN cells (Figure 1—figure supplement 1E and F). Compared to mouse cortex, fully reprogrammed iN cells (BAM 22d) show a less pronounced mCH accumulation in long genes.

“Lastly, we examined the pattern of mCH at long genes in iN cells. It was recently found that long genes are associated with greater levels of mCH in the mouse brain (Gabel et al. 2015). Comparing fully programmed iN cells to mouse cortex we found a less pronounced increase in mCH level associated with gene length in iN cells (Figure 1—figure supplement 1E and F).”

8) In the sentence "To explore the interaction between CG methylation and TF binding, in particular the binding of ASCL1, BRN2 and MYT1L, we identified 10,075 and 15,093 differentially methylated sites (DMSs) showing reduced mCG during Ascl1-only and BAM induced reprogramming, respectively (Figure 4A, C and Supplementary file 3)." 1) The numbers of DMSs 10,075 and 15,093, don't match the numbers that are summed up from the groups in the Figure 4A, C. 2) In the figure legend the information about which pair of iN cells were compared in the determination of DMSs is missing. For Ascl1-only, is it 22d vs. 2d? And for BAM-only, 22d vs. 5d? Finally, for those exact locations, were mCG estimations for 5d and MEF plotted? 3) "only 0.6% (99/15,093) BAM DMSs were overlapped with ASCL1 peaks (Figure 4D)". I suggest to verify if the percentage indicated here is correct.

We apologize for the confusion caused by the inconsistent DMS numbers in the text and figure. We excluded small DMS clusters including those less than 100 DMS, which likely reflect noise generated by setting of statistical thresholds. We have now clearly indicated this in the legend of Figure 4.

DMSs described in Figure 4 were identified and combined from all pairwise comparisons, followed by grouping by the kinetics of mCG remodeling. We have revised the legend of Figure 4 to include this information.

“Figure 4. Ascl1 and BAM factors induce distinct CG methylation reconfiguration. […] The plots show clusters containing more than 100 DMSs, which are statistically more robust.”

We have also expanded the Materials and methods section about DMS calling, and grouping as below:

“Ascl1 DMSs were identified by pairwise comparison between MEF and all Ascl1 iN cell samples. […] BAM DMSs were similarly identified, combined and grouped between MEF and all BAM iN cell samples.”

We have validated the accuracy of the number of overlaps between BAM DMSs and ASCL1 peaks. The revised manuscript now shows a more accurate percentage.

“Surprisingly, however, only 0.66% (99/15,093) of BAM DMSs overlapped with ASCL1 peaks (Figure 4D and Figure 4—figure supplement 1C).”

9) In the sentence "Among the sites that become demethylated in Ascl1-only reprogramming we found a striking co-enrichment of ASCL1 binding, in particular at early demethylating sites (Figure 4B and Figure 4—figure supplement 1C)". The ChIP-seq data for Ascl1 in MEFs show Ascl1 binding in MEFs preceding direct reprogramming, which is surprising. Same as the ChIP-seq data for BRN2 that show binding in MEFs preceding direct reprogramming. I suggest to add a brief sentence to justify this observation based on prior data, so to clarify the result.

We agree with the reviewer that adding more details of our previous study (Wapinski et al., 2013) would improve the clarity of the results. Ascl1 MEF and Brn2 MEF ChIP-seq data were generated by over-expressing Ascl1 or Brn2 individually in MEF cells for two days before the ChIP assays were performed.

“To understand the relationship between DMSs and the binding of TFs that drive the reprogramming of fibroblasts to neurons, we overlapped DMSs and ChIP-seq peaks of ASCL1, BRN2 and MYTL1 requiring a minimum of one base overlap. […] Similarly, DMSs were compared to BRN2 peaks following 2 days of overexpression of Brn2 alone (Brn2 MEF peaks), or with Ascl1 and Myt1l (Brn2 BAM peaks) in MEF (Figure 4B and D) (Wapinski et al. 2013).”

10) "we found only 0.3% (38/15,093) BAM DMSs overlapped with MYT1L binding sites" I suggest to verify if the percentage indicated here is correct. The DMSs reported in the Supplementary file 3A, B are specific CG sites 1bp. It would help to understand the data if the authors could elaborate on how the overlapping analysis was done: given the information provided, it seems that between the genomic coordinates of ChIP-seq peaks and the genomic coordinates of a specific CG site differentially methylated. Same for the motif enrichment analysis: I would add a brief sentence explaining how it was done because in the text the authors mention regions "To further illuminate the nature of the demethylating regions" Are these UMR/LMRs?

We have verified the accuracy of the overlap between BAM DMSs and MYTL1 ChIP-seq peaks.

A short description about the method used for overlapping DMSs and ChIP-seq peaks has been added to the revised manuscript:

“To understand the relationship between DMSs and the binding of TFs that drive the reprogramming of fibroblasts to neurons, we overlapped DMSs and ChIP-seq peaks of ASCL1, BRN2 and MYTL1 requiring a minimum of one base overlap.”

A short of description of motif enrichment analysis has been added to the revised manuscript:

“To further illuminate the nature of the demethylated regions we performed motif enrichment analysis for the surrounding 500bp (+/- 250bp) regions of each group of DMSs shown in Figure 4A and C.”

We further expanded the Materials and methods section regarding the method section about TF motif enrichment analysis:

“TF binding motif enrichment analysis: TF binding position weight matrices (PWM) were obtained from the MEME motif database and scanned across the mouse mm9 reference genome to identify hits using FIMO (--output-pthresh 1E-5, -- max-stored-scores 500000 and --max-strand) (Bailey et al., 2009; Grant et al., 2011). […] A TF binding motif is considered significantly enriched or depleted if the hypergeometric test resulted in q value < 1E-5.”

11) The authors state "Our results collectively suggest that mCH facilitates neuronal maturation during development through gene expression". Is it through gene expression or gene repression here?

We thank the reviewer for noticing this error. We have corrected the error in the revised manuscript.

“Our results collectively suggest that mCH facilitates neuronal maturation during development through gene repression.”

12) Optional: In Figure 1B it would be good to include the cortex MethylC-seq data from Hon et al., 2013 in order to examine if BAM 22d iN cells also cluster closely to cortex tissue.

As suggested by the reviewer, a cortex sample from Hon et al., 2013 has been added to the analyses in Figure 1B-D. As expected, BAM 22d iN cells show overall similar mCH pattern as the cortex sample.